# ReZero: Boosting MCTS-based Algorithms by Backward-view and Entire-buffer Reanalyze

## Abstract

Monte Carlo Tree Search (MCTS)-based algorithms, such as MuZero and its derivatives, have achieved widespread success in various decision-making domains. These algorithms employ the *reanalyze* process to enhance sample efficiency from stale data, albeit at the expense of significant wall-clock time consumption. To address this issue, we propose a general approach named ReZero to boost tree search operations for MCTS-based algorithms. Specifically, drawing inspiration from the one-armed bandit model, we reanalyze training samples through a backward-view reuse technique which uses the value estimation of a certain child node to save the corresponding sub-tree search time. To further adapt to this design, we periodically reanalyze the entire buffer instead of frequently reanalyzing the mini-batch. The synergy of these two designs can significantly reduce the search cost and meanwhile guarantee or even improve performance, simplifying both data collecting and reanalyzing. Experiments conducted on Atari environments, DMControl suites and board games demonstrate that ReZero substantially improves training speed while maintaining high sample efficiency.

## 1 Introduction

As a pivotal subset of artificial intelligence, Reinforcement Learning (RL) (Sutton & Barto, 1988) has acquired achievements and applications across diverse fields, including interactive gaming (Vinyals et al., 2019), autonomous vehicles (Li et al., 2022), and natural language processing (Rafailov et al., 2023). Despite its successes, a fundamental challenge plaguing traditional model-free RL algorithms is their low sample efficiency. These algorithms typically require a large amount of data to learn effectively, making them often infeasible in the real-world scenarios. In response to this problem, numerous model-based RL methods (Janner et al., 2019; Hafner et al., 2020) have emerged. These methods involve learning an additional model of the environment from data and utilizing the model to assist the agent's learning, thereby improving sample efficiency significantly.

Within this field, Monte Carlo Tree Search (MCTS) (Świechowski et al., 2023) has been proven to be a efficient method for utilizing models for planning. It incorporates the UCB1 algorithm (Auer et al., 2002) into the tree search process and has achieved promising results in a wide range of scenarios. Specifically, AlphaZero (Silver et al., 2017) plays a big role in combining deep reinforcement learning with MCTS, achieving notable accomplishments that can beat top-level human players. While it can only be applied to environments with perfect simulators, MuZero (Schrittwieser et al., 2019) extended the algorithm to cases without known environment models, resulting in good performances in a wider range of tasks. Following MuZero, many successor algorithms have emerged, enabling MuZero to be applied in continuous action spaces (Hubert et al., 2021), offline RL training scenarios (Schrittwieser et al., 2021), and etc. All these MCTS-based algorithms made valuable contributions to the universal applicability of the MCTS+RL paradigm.

However, the extensive tree search computations incur additional time overhead for these algorithms: During the data collection phase, the agent needs to execute MCTS to select an action every time it receives a new state. Furthermore, due to the characteristics of tree search, it is challenging to parallelize it using commonly used vectorized environments (Weng et al., 2022), further amplifying the speed disadvantage. On the other hand, during the reanalyze (Schrittwieser et al., 2021) process, in order to obtain higher-quality update targets, the latest models are used to re-run MCTS on the training mini-batch. The wall-clock time thus increases as a trade-off for high sample efficiency. The excessive cost has become a bottleneck hindering the further promotion of these algorithms.

Recently, a segment of research endeavors are directed toward mitigating the above wall-clock time overhead. On the one hand, SpeedyZero (Mei et al., 2023) diminishes algorithms' time overheads by deploying a parallel execution training pipeline; however, it demands additional computational resources. On the other hand, it remains imperative to identify methodologies that accelerate these algorithms without imposing extra demands. PTSAZero (Fu et al., 2023), for instance, compresses the search space via state abstraction, decreasing the time cost per search. In contrast, we aim to adopt a method that is orthogonal to both of the previous approaches. It does not require state space compression but directly reduces the search space through value estimation, and it does not introduce additional hardware overhead.

In this paper, we introduce ReZero, a new approach/framework designed to boost the MCTS-based algorithms. Firstly, inspired by the one-armed bandit model (Lattimore & Szepesvári, 2020), we propose a backward-view reanalyze technique that proceeds in the reverse direction of the trajectories, utilizing previously searched root values to bypass the exploration of specific child nodes, thereby saving time. Additionally, we have proven the convergence of our search mechanism based on the non-stationary bandit model, i.e., the distribution of child node visits will concentrate on the optimal node. Secondly, to better adapt to our proposed backward-view reanalyze technique, we have devised a novel pipeline that concentrates MCTS calls within the reanalyze process and periodically reanalyzes the entire buffer after a fixed number of training iterations. This entire-buffer reanalyze not only reduces the number of MCTS calls but also better leverages the speed advantages of parallelization. Skipping the search of specific child nodes can be seen as a pruning operation, which is common in different tree search settings. Reanalyze is also a common module in MCTS-based algorithms. Therefore, our algorithm design is universal and can be easily applied to the MCTS-based algorithm family. In addition, it will not bring about any overhead in computation resources. Empirical experiments (Section 5) show that our approach yields good results in both single-agent discrete-action environment (Atari (Bellemare et al., 2013)), two-player board games (Silver et al., 2016), and continuous control suites (Tunyasuvunakool et al., 2020), greatly improving the training speed while maintaining or even improving sample efficiency. Ablation experiments explore the impact of different reanalyze frequencies and the acceleration effect of backward-view reuse on a single search. The main contributions of this paper can be summarized as follows:

- We design a method to speed up a single tree search by the backward-view reanalyze technique. Theoretical support for the convergence of our proposed method is also provided.
- We propose an efficient framework with the entire-buffer reanalyze mechanism that further reduces the number of MCTS calls and enhance its parallelization, thus boosting MCTS-based algorithms.
- We conduct experiments on diverse environments and investigate ReZero through ablations.

## 2 RELATED WORK

### 2.1 MCTS-BASED ALGORITHMS

AlphaGo (Silver et al., 2016) and AlphaZero (Silver et al., 2017) combined MCTS with deep RL, achieving significant results in board games, defeating world-champion players, and revealing their remarkable capabilities. Following these achievements, MuZero (Schrittwieser et al., 2019) combines tree search with a learned value equivalent model (Grimm et al., 2020b), successfully promoting the algorithm's application to scenarios without known models, such as Atari. Subsequent to MuZero's development, several new works based on the MuZero framework emerged. EfficientZero (Ye et al., 2021) further improved MuZero's sample efficiency, using very little data for training and still achieving outstanding performance. Another works (Hubert et al., 2021; Antonoglou et al., 2021) extended the MuZero agent to environments with large action spaces and stochastic environments, further broadening the algorithm's usage scenario. MuZero Unplugged (Schrittwieser et al., 2021) applied MuZero's reanalyze operation as a policy enhancement operator and extended it to offline settings. Despite the aforementioned excellent improvement techniques, the time overhead of MCTS-based RL remains significant, which is the main problem this paper aims to address.

### 2.2 MCTS ACCELERATION

Recent research has focused on accelerating MCTS-based algorithms. Mei et al. (2023) reduces the algorithm's time overhead by designing a parallel system; however, this method requires more computational resources and involves some adjustments for large batch training. Fu et al. (2023) narrows

the search space through state abstraction, which amalgamates redundant information, thereby reducing the time cost per search. KataGo (Wu, 2019) use a naive information reuse trick. They save sub-trees of the search tree and serve as initialization for the next search. However, our proposed backward-view reanalyze technique is fundamentally different from this naive forward-view reuse and can enhance search results while saving time. To our knowledge, we are the first to enhance MCTS by reusing information in a backward-view. Our proposed approach can seamlessly integrate with various MCTS-based algorithms, many of which (Mei et al., 2023; Fu et al., 2023; Danihelka et al., 2022; Leurent & Maillard, 2020) are orthogonal to our contributions.

## 3 PRELIMINARIES

### 3.1 MUZERO

MuZero (Schrittwieser et al., 2019) is a fundamental model-based RL algorithm that incorporates a value-equivalent model (Grimm et al., 2020a) and leverages MCTS for planning within a learned latent space. The model consists of three core components: a *representation* model $h_\theta$, a *dynamics* model $g_\theta$, and a *prediction* model $f_\theta$:

$$
\begin{array}{lll}
\text{Representation:} & s_t = h_\theta(o_{t-l:t}) & \\
\text{Dynamics:} & s_{t+1}, r_t = g_\theta(s_t, a_t) & (1) \\
\text{Prediction:} & v_t, p_t = f_\theta(s_t) &
\end{array}
$$

The representation model transforms last observation sequences $o_{t-l:t}$ into a corresponding latent state $s_t$. The dynamics model processes this latent state alongside an action $a_t$, yielding the next latent state $s_{t+1}$ and an estimated reward $r_t$. Finally, the prediction model accepts a latent state and produces both the predicted policy $p_t$ and the state's value $v_t$. These outputs are instrumental in guiding the agent's action selection throughout its MCTS. Lastly the agent samples the best action $a_t$ following the searched visit count distribution. MuZero *Reanalyze* is an advanced version of the original MuZero. This variant enhances the model's accuracy by conducting a fresh MCTS on sampled states with the latest model, subsequently utilizing the refined policy from this search to update the policy targets. Such reanalysis yields targets of superior quality compared to those obtained during the initial data collection. Traditional uses of the algorithm have intertwined reanalysis with training, while we suggest a novel paradigm, advocating for the decoupling of the reanalyze process from training iterations, thus providing a more flexible and efficient methodology. Refer to the Appendix B for more details on MuZero during the training and inference phases.

### 3.2 BANDIT-VIEW TREE SEARCH

A stochastic bandit has $K$ arms, and playing each arm means sampling a reward from the corresponding distribution. For a search tree in MCTS, the root node can be seen as a bandit, with each child node as an arm. The left side of Figure 1 illustrates this idea. However, as the policy is continuously improved during the search process, the reward distribution for the arms should change over time. Therefore, UCT (Kocsis & Szepesvári, 2006) modeled the root node as a non-stationary stochastic bandit with a drift condition:

$$
\mathbb{P}(\hat{\mu}_{is} - \mu_i \geq \varepsilon) \leq \exp(-\frac{\varepsilon^2 s}{C^2}) \text{ and } \mathbb{P}(\hat{\mu}_{is} - \mu_i \leq -\varepsilon) \leq \exp(-\frac{\varepsilon^2 s}{C^2}) \tag{2}
$$

Where $\hat{\mu}_{is}$ is the average reward of the first $s$ samples of arm $i$. $\mu_i$ is the limit of $\mathbb{E}[\hat{\mu}_{is}]$ as $s$ approaches infinity, which indicates that the expectation of the node value converges. $C$ is an appropriate constant characterizes the rate of concentration.

Based on this modeling, UCT uses the bandit algorithm UCB1 (Auer et al., 2002) to select child nodes. AlphaZero inherits this concept and employs a variant formula:

$$
UCB_{score}(s, a) = Q(s, a) + cP(s, a)\frac{\sqrt{\sum_b N(s, b)}}{1 + N(s, a)} \tag{3}
$$

where $s$ is the state corresponding to the current node, $a$ is the action corresponding to a child node. $Q(s, a)$ is the mean return of choosing action $a$, $P(s, a)$ is the prior score of action $a$, $N(s, a)$ is the total time that $a$ has been chosen, $\sum_b N(s, b)$ is the total time that $s$ has been visited. Viewing tree search from the bandit-view inspired us to use techniques from the field of bandits to improve the tree search. In next section, We use the idea of the one-armed bandit model to design our algorithm and prove the convergence of our algorithm based on the non-stationary bandit model.

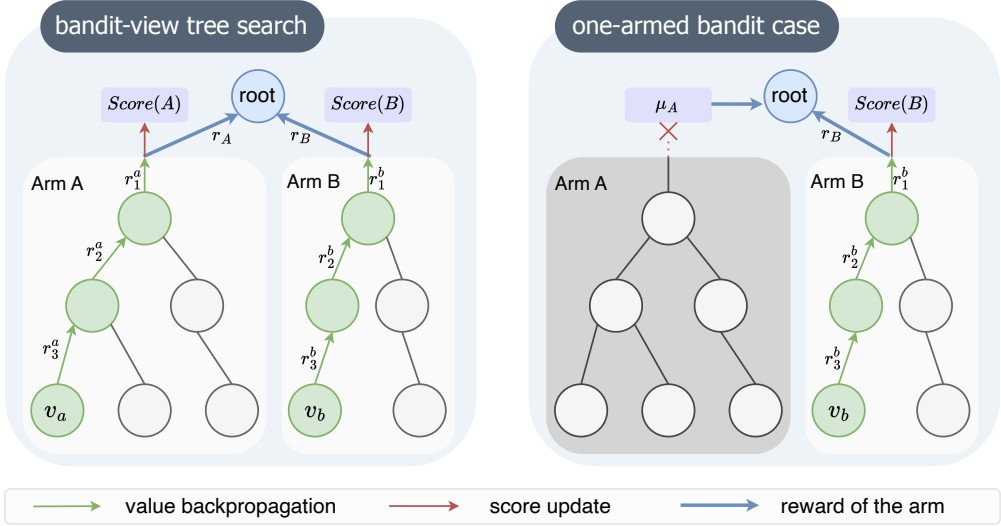

Figure 1: The connection between MCTS and bandits. **Left** shows tree search in a bandit-view. When the action $A$ is selected, a return $r_A$ will be returned, where $r_A = \sum_{t=1}^{3} \gamma^{t-1} r_t^a + \gamma^3 v_a$. For the root node, the traversal, evaluation and back-propagation occurring in the sub-tree can be approximated as sampling from an non-stationary distribution. Thus it can be seen as a non-stationary bandit. **Right** shows the one-armed bandit case. Once the true value $\mu_A$ is known, we can evaluate arm A using $\mu_A$, thereby eliminating the need to rely on subsequent tree search processes.

## 4    METHOD

In this section, we introduce the specific design of ReZero. Section 4.1 describes the inspiration and specific operations of backward-view reanalyze. It performs a reverse search on the trajectory and uses the value estimation of a certain child node to save the corresponding sub-tree search time. Section 4.2 analyzes the node selection method introduced in Section 4.1 and ultimately proves the convergence of the method. Section 4.3 introduced the overall framework of ReZero, which adopts the buffer reanalyze setting to better integrate with the method described in Section 4.1.

### 4.1    BACKWARD-VIEW REANALYZE

Our algorithm design stems from a simple inspiration: if we could know the true state-value (e.g., expected long-term return) of a child node in advance, we could save the search for it, thus conserving search time. As shown on the right side of Figure 1, we directly use the true expectation $\mu_A$ to evaluate the quality of Arm A, thereby eliminating the need for the back-propagated value to calculate the score in Eq. 3. This results in the process occurring within the gray box in Figure 1 being omitted. Indeed, this situation can be well modeled as an **one-armed bandit** (Lattimore & Szepesvári, 2020), which also facilitates our subsequent theoretical analysis.

Driven by the aforementioned motivation, we aspire to obtain the expected return of a child node in advance. However, the true value is always unknown. Consequently, we resort to using the root value obtained from MCTS as an approximate substitute. Specifically, for two adjacent time steps $S_0^1$ and $S_0^0$ [1] in Figure 2, when searching for state $S_0^1$, the root node corresponds to a child node of $S_0^0$. Therefore, the root value obtained can be utilized to assist the search for $S_0^0$. However, there is a temporal contradiction. $S_0^1$ is the successor state of $S_0^0$, yet we need to complete the search for $S_0^1$ first. Fortunately, this is possible during the reanalyze process.

During reanalyze, since the trajectories were already collected in the replay buffer, we can perform tree search in a backward-view, which searches the trajectory in a reverse order. Figure 2 illustrates a batch containing $n+1$ trajectories, each of length $k+1$, and $S_l^t$ is the $t$-th state in the $l$-th trajectory. We first conduct a search for all $S^k$s, followed by a search for all $S^{k-1}$s, and so on. After we search

---

[1] The subscript denotes the trajectory, the superscript denotes the time step in the trajectory.

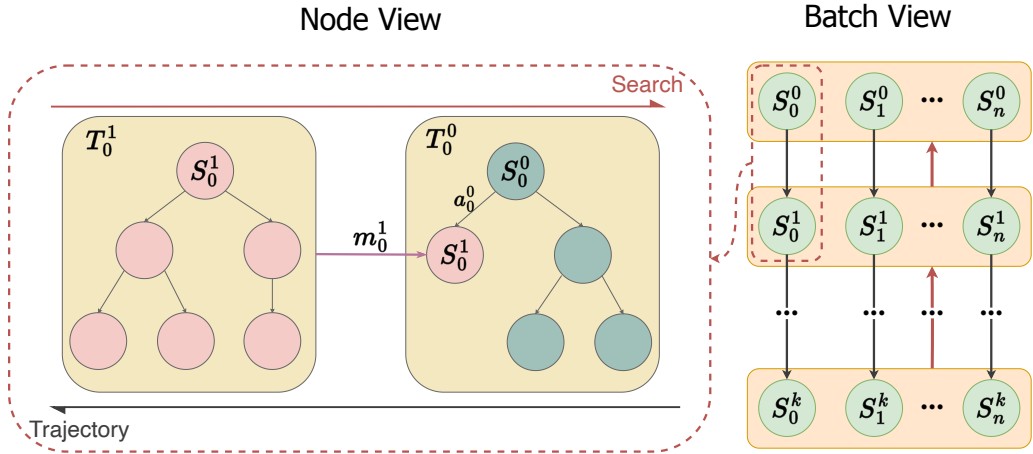

Figure 2: An illustration about the backward-view reanalyze in node and batch view. We sample $n + 1$ trajectories of length $k + 1$ to form a batch and conduct the search in the reverse direction of trajectories. From the node view, we would first search $S_0^1$ and then pass root value $m_0^1$ to $S_0^0$ to evaluate the value of a child node. $T_0^1$ and $T_0^0$ are the corresponding search trees. From the batch view, we would group all $S^1$s into a sub batch to search together and pass the root values to the $S^0$s.

on state $S_l^{t+1}$ [2], the root value $m_l^{t+1}$ is obtained. When engaging in search on $S_l^t$, we assign the value of $S_l^{t+1}$ to the fixed value $m_l^{t+1}$. During traverse in the tree, we select the action $a_{root}$ for root node $S_l^t$ with the following equation:

$$a_{root} = \arg\max_a I_l^t(a) \tag{4}$$

$$I_l^t(a) = \begin{cases} UCB_{score}(S_l^t, a), & a \neq a_l^t \\ r_l^t + \gamma m_l^{t+1}, & a = a_l^t \end{cases} \tag{5}$$

where $a$ refers to the action associated with a child node, $a_l^t$ is the action corresponding to $S_l^{t+1}$, and $r_l^t$ signifies the reward predicted by the dynamic model. If an action distinct from $a_l^t$ is selected, the simulation continues its traversal with the original setting as in MuZero. *If action $a_l^t$ is selected, this simulation is terminated immediately.* Since the time used to search for node $S_l^{t+1}$ is saved, this enhanced search process is faster than the original version. Algorithm 1 shows the specific design with Python-like code.[3]

---

**Algorithm 1** Python-like code for information reuse

```
# trajectory_segment: a segment with length K    # N: simulation numbers during one search
def search_backwards(trajectory_segment):        def reuse_MCTS(root, action, values)
    # prepare search context from the segment        for i in range(N):
    roots, actions = prepare(trajectory_segment)         # select an action for root node
    policy_targets = []                                  a = select_root_child(root, action, value):
    # search the roots backwards                         # early stop the simulation
    for i in range(K, 1, -1):                            if a == action:
        if i == K:                                           backpropagate()
            # origin MCTS for Kth root                       break
            policy, value = origin_MCTS(roots[i])        # traverse to the leaf node
        else:                                            else:
            # reuse information from previous sear           traverse()
            policy, value = reuse_MCTS(                      backpropagate()
                roots[i], actions[i], value
            )
        policy_targets.append(policy)
```

---

[2]To facilitate the explanation, we have omitted details such as the latent space and do not make a deliberate distinction between states and nodes.

[3]Our enhanced search process is implemented through $reuse\_MCTS()$, where $select\_root\_child()$ performs the action selection method of Equation 5. The $traverse()$ and $backpropagate()$ represent the forward search and backward propagation processes in standard MCTS.

## 4.2 THEORETICAL ANALYSIS

AlphaZero selects child nodes using Equation 3 and takes the final action based on the visit counts. In the previous section, we replace Equation 3 with Equation 5, which undoubtedly impacts the visit distribution of child nodes. Additionally, since we use the root value as an approximation to true expectation, the error between the two may also affect the search results. To demonstrate the reliability of our algorithm, in this section, we model the root node as an non-stationary bandit and prove that, as the number of total visit increases, the visit distribution gradually concentrates on the optimal arm. Specifically, we have the following theorem:

**Theorem 1**  For a non-stationary bandit that conforms to the assumptions of Equation 2, denote the total number of rounds as $n$, the prior score for arm $i$ as $P_i$, and the number of times a sub-optimal arm $i$ is selected in $n$ rounds as $T_i(n)$, then use a sampled estimation instead of UCB value to evaluate a specific arm(like we do in Equation 5) can ensure that $\frac{\mathbb{E}[T_i(n)]}{n} \to 0$ as $n \to \infty$. Specifically, if we know the $n$ times sample mean $\hat{\mu}^*$ of the optimal arm in advance, then $\mathbb{E}[T_i(n)]$ for all sub-optimal arm $i$ satisfies

$$\mathbb{E}[T_i(n)] \le 2 + \frac{2P_i\sqrt{n-1}}{\Delta_i - \varepsilon} + \frac{C^2}{(\Delta_i - \varepsilon)^2} + n\exp\left(-\frac{n\varepsilon^2}{C^2}\right) \tag{6}$$

Otherwise, if we have the $n$ times sample mean $\hat{\mu}_l$ of a sub-optimal arm $l$, then for arm $l$,

$$\mathbb{E}[T_l(n)] \le 1 + \frac{2C^6}{\varepsilon^4 P_1^2} + n\exp\left(-\frac{n(\Delta_l - \varepsilon)^2}{C^2}\right) \tag{7}$$

and for other sub-optimal arms,

$$\mathbb{E}[T_i(n)] \le 3 + \frac{2P_i\sqrt{n-1}}{\Delta_i - \varepsilon} + \frac{C^2}{(\Delta_i - \varepsilon)^2} + \frac{2C^6}{\varepsilon^4 P_1^2} \tag{8}$$

where $\Delta_i$ is the optimal gap for arm $i$, $\varepsilon$ is a constant in $(0, \Delta_i)$, and $C$ is the constant in Eq. 2.

We provide the complete proof and draw similar conclusions for AlphaZero in Appendix A. Additionally, our method has a lower upper bound for $\mathbb{E}[T_i(n)]$. This implies that our algorithm may yield visit distribution more concentrated on the optimal arm. This is a potential worth exploring in future work, especially in offline scenarios (Schrittwieser et al., 2021) where reanalyze becomes the sole method for policy improvement.

## 4.3 THE REZERO FRAMEWORK

The technique introduced in Section 4.1 will bring a new problem in practice. The experiments in the Appendix D.3 demonstrate that batching MCTS allows for parallel model inference and data processing, thereby accelerating the average search speed. However, as shown in the Figure 2, to conduct the backward-view reanalyze, we need to divide the batch into $\frac{1}{k+1}$ of its original size. This diminishes the benefits of parallelized search and, on the contrary, makes the algorithm slower.

We propose a new pipeline that is more compatible with the method introduced in Section 4.1. In particular, during the collect phase, we have transitioned from using MCTS to select actions to directly sampling actions based on the policy network's output. This shift can be interpreted as an alternative method to augment exploration in the collect phase, akin to the noise injection at the root node in MCTS-based algorithms. Figure 11 in Appendix D.3 indicates that this modification does not significantly compromise performance during evaluation. For the reanalyze process, we introduce the periodical entire-buffer reanalyze. As shown in Figure 3, we reanalyze the whole buffer after a fixed number of training iterations. For each iteration, we do not need to run MCTS to reanalyze the mini-batch, only need to sample the mini-batch and execute the gradient descent.

Overall, this design offers two significant advantages: ① The entire buffer reanalyze is akin to the fixed target net mechanism in DQN (Mnih et al., 2013), maintaining a constant policy target for a certain number of training iterations. Reducing the frequency of policy target updates correspondingly decreases the number of MCTS calls. Concurrently, this does not result in a decrease in performance. ② Since we no longer invoke MCTS during the collect phase, all MCTS calls

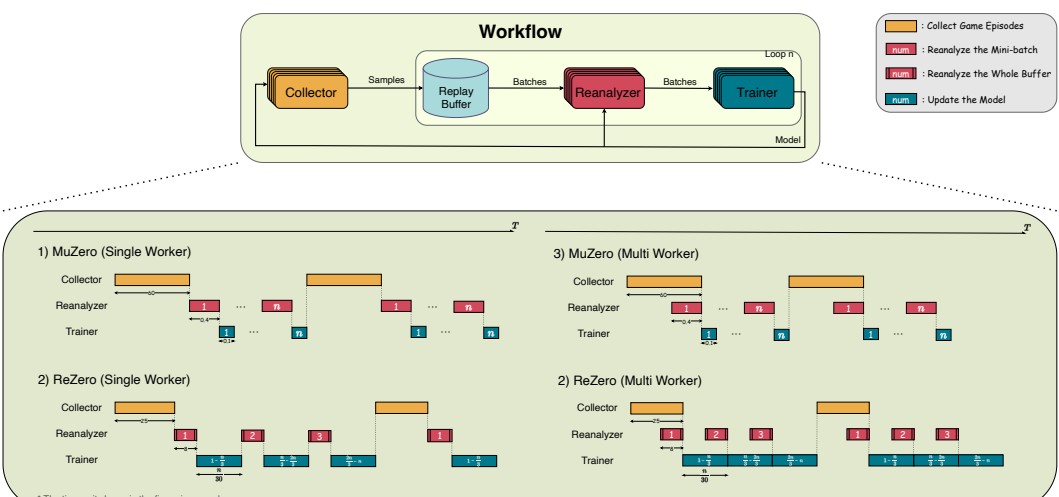

Figure 3: Execution workflow and runtime cycle graph about MuZero and ReZero in both single and multiple worker cases. The number inside the modules represent the number of iterations, and the number under the modules represent the time required for module execution. The model is updated $n$ iterations between two collections. MuZero reanalyzes the mini-batch before each model update. ReZero reanalyzes the entire buffer after certain iterations($\frac{n}{3}$ for example), which not only reduces the total number of MCTS calls, but also takes advantage of the processing speed of large batches.

are concentrated in the reanalyze process. And during the entire-buffer reanalyze, we are no longer constrained by the size of the mini-batch, allowing us to freely adjust the batch size to leverage the advantages of large batches. Figure 12 shows both excessively large and excessively small batch sizes can lead to a decrease in search speed. We choose the batch size of 2000 according to the experiment in Appendix D.3.

Experiments show that our algorithm maintains high sample efficiency and greatly save the running time of the algorithm. Our pipeline also has the following potential improvement directions:

- When directly using policy for data collection, action selection is no longer bound by tree search. Thus, previous vectorized environments like Weng et al. (2022) can be seamlessly integrated. Besides, this design makes MCTS-based algorithms compatible with existing RL exploration methods like Badia et al. (2020).

- Our method no longer needs to reanalyze the mini-batch for each iteration, thus decoupling the process of *reanalyze* and *training*. This provides greater scope for parallelization. In the case of multiple workers, we can design efficient parallelization paradigms as shown in Figure 3.

- We can use a more reasonable way, such as weighted sampling to preferentially reanalyze a part of the samples in the buffer, instead of simply reanalyzing all samples in the entire buffer. This is helpful to further reduce the computational overhead.

## 5 EXPERIMENT

*Efficiency* in RL usually refers to two aspects: *sample efficiency*, the agent's ability to learn effectively from a limited number of environmental interactions, gauged by the samples needed to reach a successful policy using equal compute resources; *time efficiency*, which is how swiftly an algorithm learns to make optimal decisions, indicated by the wall-clock time taken to achieve a successful policy. **Our primary goal is to improve time efficiency without compromising sample efficiency**. Here we first give a toy example case in Section 5.1 to intuitively demonstrate the acceleration effect of our design, and the corresponding code example is placed in the Appendix D.4 for readers to quickly understand the algorithm design.

And then, to validate the efficiency of our introduced two boosting techniques for MCTS-based algorithms, we have incorporated the ReZero with two prominent algorithms detailed in Niu et al. (2023): MuZero with a Self-Supervised Learning loss (SSL) and EfficientZero, yielding the enhanced vari-

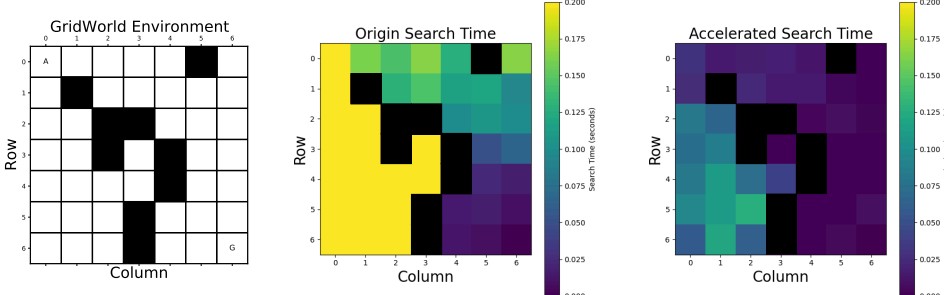

Figure 4: Acceleration effect on the toy case. **Left** is a simple maze environment where the agent starts at point A and receives a reward of size 1 upon reaching the end point G. **Middle** shows the search time corresponding to each position when set as the root node. Meanwhile, the root node values obtained during the search are preserved. **Right** shows the corresponding search time when these root node values are used to assist the search. The comparison shows that the search duration is generally reduced. For specific experimental settings and code, please refer to the Appendix D.4.

ants ReZero-M and ReZero-E respectively. For brevity, when the context is unambiguous, we simply refer to them as ReZero, omitting the symbol representing the baseline. It is important to highlight that ReZero can be seamlessly integrated with various algorithm variants of MuZero. In this context, we have chosen to exemplify this integration using the above two instances. In order to validate the applicability of our method across various decision-making environments, we opt for 26 representative *Atari* environments characterized by classic image-input observation and discrete action spaces, in addition to the strategic board games *Connect4* and *Gomoku* with special state spaces, and two continuous control tasks in *DMControl* (Tunyasuvunakool et al., 2020). For a baseline comparison, we employed the original implementations of MuZero and EfficientZero as delineated in the LightZero benchmark (Niu et al., 2023). The full implementation details available in Appendix C. Besides, we emphasize that to ensure a fair comparison of wall-clock time, all experimental trials were executed on a fixed single worker hardware settings. In Section 5.2, Section 5.3 and Section 5.4, we sequentially explore and try to answer the following three questions:

- How much can ReZero-M/ReZero-E improve time efficiency compared to MuZero/EfficientZero, while maintaining equivalent levels of high sample efficiency?

- What is the effect and hyper-parameter sensitivity of the *Entire-buffer Reanalyze* technique?

- How much search budgets can be saved in practice by the *Backward-view Reanalyze* technique?

## 5.1 TOY CASE

Intuitively, it is evident that our proposed method can achieve a speed gain because we eliminate the search of a certain subtree, especially when this subtree corresponds to the optimal action (which often implies that the subtree has a larger number of nodes). We conduct an experiment on a toy example case and include the experimental code in the Appendix D.4. This helps to visually illustrate the speed gain achieved by skipping subtree search and allows readers to quickly understand the algorithm design through simple code. As shown in Figure 4 (Left), we implement a simple $7 \times 7$ maze environment where the agent starts at point A and receives a reward of 1 upon reaching point G. We perform an MCTS with each position in the maze as the root node and recorded the search time in Figure 4 (Middle). It can be seen that regions farther from the end point require more time to search (this is related to our simulation settings, see the appendix for details). For comparision, we also performed searches with each position as the root node, but during the search, we used the root node values obtained from the search in Figure 4 (Middle) to evaluate specific actions. The experimental time was recorded in Figure 4 (Right). It can be seen that after eliminating the search of specific subtrees, the search time was generally reduced. This simple result validates the rationality of our algorithm design. In the next section, we will specifically validate the time efficiency of the ReZero framework in diverse decision-making tasks in Section 5.2.

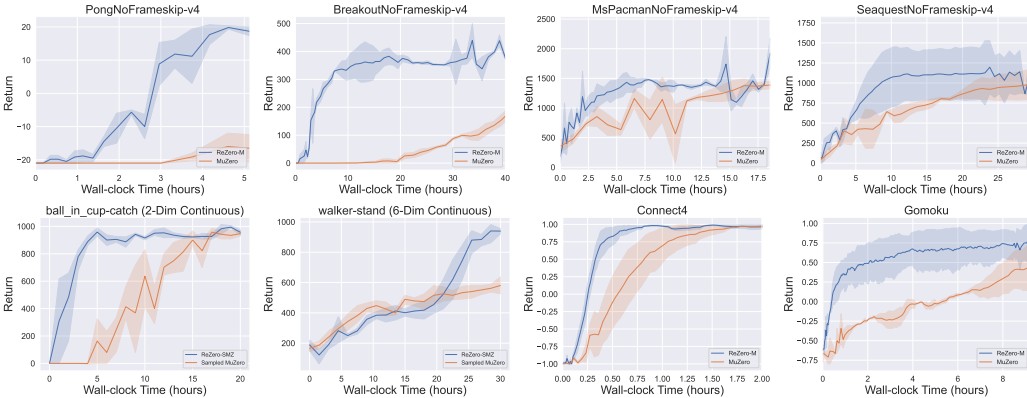

Figure 5: **Time-efficiency** of ReZero-M vs. MuZero on four representative *Atari* games, two continuous control tasks of *DMControl* (*ball_in_cup-catch*, *walker-stand*), and two board games (*Connect4*, *Gomoku*). The horizontal axis represents *Wall-clock Time* (hours), while the vertical axis indicates the *Episode Return* over 5 evaluation episodes. ReZero-M demonstrates superior time-efficiency compared to the baseline across a diverse set of games, encompassing both image and state observations, discrete and continuous actions, and scenarios involving sparse rewards. These figures compute mean of 5 runs, and shaded areas are 95% confidence intervals.

| | | Atari | | | DMControl | | Board Games | |
|---|---|---|---|---|---|---|---|---|
| *avg. wall time (h) to 100k env. steps ↓* | Pong | Breakout | MsPacman | Seaquest | ball_in_cup-catch | walker-stand | Connet4 | Gomoku |
| **ReZero-M** (ours) | $1.0_{\pm 0.1}$ | $3.0_{\pm 0.8}$ | $1.4_{\pm 0.2}$ | $1.9_{\pm 0.4}$ | $2.1_{\pm 0.2}$ | $4.3_{\pm 0.3}$ | $5.5_{\pm 0.6}$ | $4.5_{\pm 0.5}$ |
| MuZero (Schrittwieser et al., 2019) | $4.0_{\pm 0.5}$ | $4.9_{\pm 1.8}$ | $6.9_{\pm 0.3}$ | $10.1_{\pm 0.5}$ | $5.6_{\pm 0.4}$ | $9.5_{\pm 0.6}$ | $9.1_{\pm 0.8}$ | $15.3_{\pm 1.5}$ |

Table 1: **Average wall-time** of ReZero-M vs. MuZero on various tasks. *(left)* Four Atari games, *(middle)* two control tasks, *(right)* two board games. The time represents the average total wall-time to 100k environment steps for each algorithm. Mean and standard deviation over 5 runs.

## 5.2 TIME EFFICIENCY

**Setup:** We aim to evaluate the performance of ReZero against classical MCTS-based algorithms MuZero and EfficientZero, focusing on the wall-clock time reduction required to achieve the comparable performance level. In order to facilitate a fair comparison in terms of wall-time, we not only utilize identical computational resources but also maintain consistent hyper-parameter settings across the algorithms (unless specified cases). Key parameters are aligned with those from original papers. After each data collection phase, the model is trained for multiple iterations according to the replay ratio, which is denoted as one training epoch. Specifically, we set the *replay ratio* (the ratio between environment steps and training steps) (D'Oro et al., 2022) to 0.25, and the *reanalyze ratio* (the ratio between targets computed from the environment and by reanalysing existing data) (Schrittwieser et al., 2021) is set to 1. For detailed hyper-paramater configurations, please refer to the Appendix C.

**Results:** Our experiments shown in Figure 5 illustrates the training curves and performance comparisons in terms of wall-clock time between ReZero-M and the MuZero across various decision-making environments. Note that on continuous control tasks we use the sampled version of ReZero-M and MuZero Hubert et al. (2021). The data clearly indicates that ReZero-M achieves a significant improvement in time efficiency, attaining a near-optimal policy in significantly less time for these eight diverse decision-making tasks. To provide an alternative perspective on the time cost of training on the same number of environment steps, we also provide an in-depth comparison of the wall-clock training time up to 100k environment steps for ReZero-M against MuZero in Table 1. For a more comprehensive understanding of our findings, we offer the complete table of the 26 Atari games usually used for sample-efficient RL in Appendix D.1. It reveals that, on most games, ReZero require between 2-4 times less wall-clock time per 100k steps compared to the baselines, while maintaining comparable or even superior performance in terms of episode return. Additional results about another algorithm comparison instance ReZero-E and EfficientZero are detailed in Appendix D.2, specifically in Figure 8 and 9. This setting also supports the time efficiency of our proposed ReZero framework.

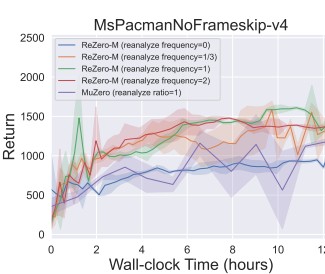 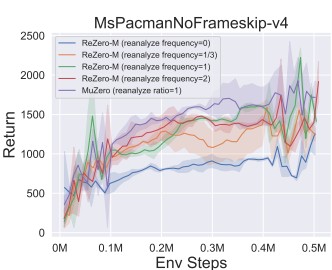

Figure 6: The ablation experiment of **Reanalyze Frequency** in ReZero-M on the *Atari MsPacman* game. The proper reanalyze frequency can improve time and sample efficiency while obtaining the comparable return with MuZero (*reanalyze ratio*=1).

## 5.3 EFFECT OF REANALYZE FREQUENCY

In next two sub-sectons, we will delve into details to explain how ReZero works. Firstly, we will analyze how the entire-buffer reanalyze technique manages to simplify the original iterative mini-batch reanalyze scheme to the periodical updated version while maintaining high sample efficiency.

Here, we adjust the periodic *reanalyze frequency*—which determines how often the buffer is reanalyzed during a training epoch—in ReZero for the *MsPacman* environment. Specifically, we set *reanalyze frequency* to $\{0, \frac{1}{3}, 1, 2\}$. The original MuZero variants with the *reanalyze ratio* of 1 is also included in this ablation experiment as a baseline. Figure 6 shows entire training curves in terms of *Wall-time* or *Env Steps* and validates that appropriate reanalyze frequency can save the time overhead without causing any obvious performance loss.

## 5.4 EFFECT OF BACKWARD-VIEW REANALYZE

| Indicators | Avg.time (ms) | tree search (num calls) | dynamics (num calls) | data process (num calls) |
|---|---|---|---|---|
| **ReZero-M** | **0.69 ± 0.02** | **6089** | **122** | **277** |
| MuZero | 1.08 ± 0.09 | 13284 | 256 | 455 |

Table 2: Comparisons about the detailed time cost indicators between MuZero and ReZero-M inside the tree search.

To further validate and understand the advantages and significance of the backward-view reanalyze technique proposed in ReZero, we meticulously document a suite of statistical indicators of the tree search process in Table 2. The number of function calls is the cumulative value of 100 training iterations on *Pong*. *Avg. time* is the average time of a MCTS across all calls. Comparative analysis between ReZero-M and MuZero reveals that the backward-view reanalyze technique reduces the invocation frequency of the dynamics model, the search tree, and other operations like data process transformations. Consequently, this advanced technique in leveraging subsequent time step data contributes to save the tree search time in various MCTS-based algorithms, further leading to overall wall-clock time gains. The complexity and implementation of the tree search process directly influences the efficiency gains achieved through the backward-view reanalyze technique. As the tree search becomes more intricate and sophisticated, e.g. Sampled MuZero (Hubert et al., 2021), the time savings realized through this method are correspondingly amplified. Additionally, Theorem 1 demonstrates that the backward-view reanalyze can reduce the regret upper bound, which indicates a better search result. Besides, Figure 10 in Appendix D.3 shows a comparison of sample efficiency between using backward-view reanalyze and origin reanalyze process in *MsPacman*. The experimental results reveal that our method not only enhances the speed of individual searches but also improves sample efficiency. This aligns with the theoretical analysis.

## 6 CONCLUSION AND LIMITATION

In this paper, we have delved into the efficiency and scalability of MCTS-based algorithms. Unlike most existing works, we incorporate information reuse and periodic reanalyze techniques to reduces wall-clock time costs while preserving sample efficiency. Our theoretical analysis and experimental results confirm that ReZero efficiently reduces the time cost and maintains or even improves performance across different decision-making domains. However, our current experiments are mainly conducted on the single worker setting, there exists considerable optimization scope to apply our approach into distributed RL training, and our design harbors the potential of better parallel acceleration and more stable convergence in large-scale training tasks. Also, the combination between

ReZero and frameworks akin to AlphaZero, or its integration with some recent offline datasets such as RT-X (Padalkar et al., 2023), constitutes a fertile avenue for future research. These explorations could broaden the application horizons of MCTS-based algorithms. Additionally, since in offline training scenarios, reanalyze becomes the only means of policy improvement, this makes the acceleration of the reanalyze phase in ReZero even more critical. Moreover, the potential improvement in search results by ReZero may further improve the training result, rather than merely accelerating the training. Therefore, combining ReZero with MuZero Unplugged (Schrittwieser et al., 2021) is a direction worth exploring for building foundation models for decision-making.

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

# A PROOF MATERIALS

In this section, we will present the complete supplementary proofs.

Lemma A: Let $w_t$ be a random variable that satisfies the concentration condition in Equation 2 with zero expectation, $\varepsilon > 0$, $a > 0$ and

$$\kappa = \sum_{t=1}^{n} \mathbb{I}\{w_t + \sqrt{\frac{a}{t^2}} \geq \varepsilon\} \tag{9}$$

then it holds that $E[\kappa] \leq 1 + \frac{2\sqrt{a}}{\varepsilon} + \frac{C^2}{\varepsilon^2}$.

*Proof.* Take $u$ as $\frac{2\sqrt{a}}{\varepsilon}$, then

$$E[\kappa] \leq u + \sum_{t=\lceil u \rceil}^{n} \mathbb{P}(w_t + \sqrt{\frac{a}{t^2}} \geq \varepsilon) \tag{10}$$

$$\leq u + \sum_{t=\lceil u \rceil}^{n} \exp(-\frac{t(\varepsilon - \sqrt{\frac{a}{t^2}})^2}{C^2}) \tag{11}$$

$$\leq 1 + u + \int_{u}^{\infty} \exp(-\frac{t(\varepsilon - \sqrt{\frac{a}{t^2}})^2}{C^2})dt \tag{12}$$

$$\leq 1 + u + e^{2\frac{\sqrt{a}\varepsilon}{C^2}} \int_{u}^{\infty} e^{-\frac{t\varepsilon^2}{C^2}} dt \tag{13}$$

$$= 1 + u + \frac{C^2}{\varepsilon^2} = 1 + \frac{2\sqrt{a}}{\varepsilon} + \frac{C^2}{\varepsilon^2} \tag{14}$$

$\square$

**Proof for Theorem 1:**

*Proof.* We first present the upper bound of $\mathbb{E}[T_i(n)]$ for using the Equation 3 in AlphaZero. A slight adjustment to this proof yields the conclusion of Theorem 1. Denote $T_i(k)$ as the number of times that arm $i$ has been chosen until time $k$, $A_t$ as the arm selected at time $t$, $\hat{\mu}_{is}$ as the average of the first $s$ samples of arm $i$, $\mu_i$ as the limit of $\mathbb{E}[\hat{\mu}_{is}]$, which satisfies the concentration assumption in Equation 2, and $\hat{\mu}_i(k) = \hat{\mu}_{iT_i(k)}$. Without loss of generality, we assume that arm 1 is the optimal arm. Then we have:

$$T_i(n) = \sum_{t=1}^{n} \mathbb{I}\{A_t = i\} \tag{15}$$

$$\leq \sum_{t=1}^{n} \mathbb{I}\{\hat{\mu}_1(t-1) + P_1\frac{\sqrt{t-1}}{1+T_1(t-1)} \leq \mu_1 - \varepsilon\} \tag{16}$$

$$+ \sum_{t=1}^{n} \mathbb{I}\{\hat{\mu}_i(t-1) + P_i\frac{\sqrt{t-1}}{1+T_i(t-1)} \geq \mu_1 - \varepsilon \text{ and } A_t = i\} \tag{17}$$

for Equation 16, we have

$$\mathbb{E}[\sum_{t=1}^{n} \mathbb{I}\{\hat{\mu}_1(t-1) + P_1 \frac{\sqrt{t-1}}{1+T_1(t-1)} \leq \mu_1 - \varepsilon\}] \tag{18}$$

$$\leq 1 + \sum_{t=2}^{n} \sum_{s=0}^{t-1} \mathbb{P}(\hat{\mu}_{1s} + P_1 \frac{\sqrt{t-1}}{1+s} \leq \mu_1 - \varepsilon) \tag{19}$$

$$= 1 + \sum_{t=2}^{n} \sum_{s=1}^{t-1} \mathbb{P}(\hat{\mu}_{1s} + P_1 \frac{\sqrt{t-1}}{1+s} \leq \mu_1 - \varepsilon) \tag{20}$$

$$\leq 1 + \sum_{t=2}^{n} \sum_{s=1}^{t-1} \exp(-\frac{s(\varepsilon + P_1 \frac{\sqrt{t-1}}{1+s})^2}{C^2}) \tag{21}$$

$$\leq 1 + \sum_{t=2}^{n} \exp(-\frac{1}{C^2}\varepsilon P_1 \sqrt{t-1}) \sum_{s=1}^{t-1} \exp(-\frac{s\varepsilon^2}{C^2}) \tag{22}$$

$$\leq 1 + \sum_{t=2}^{n} \exp(-\frac{1}{C^2}\varepsilon P_1 \sqrt{t-1}) \frac{C^2}{\varepsilon^2} \tag{23}$$

$$\leq 1 + \frac{2C^6}{\varepsilon^4 P_1^2} \tag{24}$$

Notes: In Equation 19, since we assume $P_1 \geq \mu_1$, the probability of the term $s = 0$ would be 0. Thus, we can discard it. If this assumption doesn't hold, we can choose to accumulate $t$ starting from a larger $t_0$(which satisfies $P_1\sqrt{t_0 - 1} > \mu_1$ as mentioned in the article). Starting the summation from such a $t_0$ ensures all terms of $s = 0$ can still be discarded, and all add terms that $t \leq t_0$ can be bounded to 1. This only changes the constant term and won't affect the growth rate of regret. From Equation 21 to Equation 22, we just need to expand the quadratic term and do some simple inequality scaling. From Equation 22 to Equation 23, we need to notice that $\sum_{s=1}^{t-1} \exp(-\frac{s\varepsilon^2}{C^2})$ is a geometric sequence and scale it to $\frac{C^2}{\varepsilon^2}$. From Equation 23 to Equation 24, we use the inequality $\sum_{t=2}^{n} \frac{1}{e^{a\sqrt{t-1}}} \leq \int_1^{\infty} \frac{1}{e^{a\sqrt{t-1}}}$.

And for Equation 17, we have

$$\mathbb{E}[\sum_{t=1}^{n} \mathbb{I}\{\hat{\mu}_i(t-1) + P_i \frac{\sqrt{t-1}}{1+T_i(t-1)} \geq \mu_1 - \varepsilon \text{ and } A_t = i\}] \tag{25}$$

$$\leq \mathbb{E}[\sum_{t=1}^{n} \mathbb{I}\{\hat{\mu}_i(t-1) + P_i \sqrt{\frac{n-1}{(1+T_i(t-1))^2}} \geq \mu_1 - \varepsilon \text{ and } A_t = i\}] \tag{26}$$

$$\leq 1 + \mathbb{E}[\sum_{s=1}^{n-1} \mathbb{I}\{\hat{\mu}_{is} + P_i \sqrt{\frac{n-1}{(1+s)^2}} \geq \mu_1 - \varepsilon\}] \tag{27}$$

$$\leq 1 + \mathbb{E}[\sum_{s=1}^{n} \mathbb{I}\{\hat{\mu}_{is} - \mu_i + P_i \sqrt{\frac{n-1}{s^2}} \geq \Delta_i - \varepsilon\}] \tag{28}$$

with Lemma A, we can have

$$28 \leq 2 + \frac{2P_i\sqrt{n-1}}{\Delta_i - \varepsilon} + \frac{C^2}{(\Delta_i - \varepsilon)^2} \tag{29}$$

so we have

$$\mathbb{E}[T_i(n)] \leq 3 + \frac{2P_i\sqrt{n-1}}{\Delta_i - \varepsilon} + \frac{C^2}{(\Delta_i - \varepsilon)^2} + \frac{2C^6}{\varepsilon^4 P_1^2} \tag{30}$$

The proof of theorem 1 can be obtained by making slight modifications. In case we have drawn $n$ samples from the same non-stationary distribution as arm 1, and the average of these first $n$ samples is $\hat{\mu}_1$,

$$\mathbb{E}[T_i(n)] = \mathbb{E}[\sum_{t=1}^{n} \mathbb{I}\{A_t = i\}] \tag{31}$$

$$\leq \mathbb{E}[\sum_{t=1}^{n} \mathbb{I}\{\hat{\mu}_1 \leq \mu_1 - \varepsilon\}] \tag{32}$$

$$+ \mathbb{E}[\sum_{t=1}^{n} \mathbb{I}\{\hat{\mu}_i(t-1) + P_i \frac{\sqrt{t-1}}{1+T_i(t-1)} \geq \mu_1 - \varepsilon \text{ and } A_t = i\}] \tag{33}$$

$$\leq n \exp\left(-\frac{n\varepsilon^2}{C^2}\right) + \mathbb{E}[\sum_{t=1}^{n} \mathbb{I}\{\hat{\mu}_i(t-1) + P_i \frac{\sqrt{t-1}}{1+T_i(t-1)} \geq \mu_1 - \varepsilon \text{ and } A_t = i\}] \tag{34}$$

$$\leq 2 + \frac{2P_i\sqrt{n-1}}{\Delta_i - \varepsilon} + \frac{C^2}{(\Delta_i - \varepsilon)^2} + n \exp\left(-\frac{n\varepsilon^2}{C^2}\right) \tag{35}$$

In case we have drawn $n$ samples from the same non-stationary distribution as arm $l$, and the average of these first $n$ samples is $\hat{\mu}_l$,

$$\mathbb{E}[T_l(n)] = \mathbb{E}[\sum_{t=1}^{n} \mathbb{I}\{A_t = l\}] \tag{36}$$

$$\leq \mathbb{E}[\sum_{t=1}^{n} \mathbb{I}\{\hat{\mu}_1(t-1) + P_1 \frac{\sqrt{t-1}}{1+T_1(t-1)} \leq \mu_1 - \varepsilon\}] \tag{37}$$

$$+ \mathbb{E}[\sum_{t=1}^{n} \mathbb{I}\{\hat{\mu}_l \geq \mu_1 - \varepsilon \text{ and } A_t = l\}] \tag{38}$$

$$\leq \mathbb{E}[\sum_{t=1}^{n} \mathbb{I}\{\hat{\mu}_1(t-1) + P_1 \frac{\sqrt{t-1}}{1+T_1(t-1)} \leq \mu_1 - \varepsilon\}] + n \exp\left(-\frac{n(\Delta_l - \varepsilon)^2}{C^2}\right) \tag{39}$$

$$\leq 1 + \frac{2C^6}{\varepsilon^4 P_1^2} + n \exp\left(-\frac{n(\Delta_l - \varepsilon)^2}{C^2}\right) \tag{40}$$

and the bound of $E[T_i(n)]$ for $i \neq l$ keeps unchanged. $\square$

## B  MuZero

During the *inference* phase, the representation model transforms a sequence of the last $l$ observations $o_{t-l:t}$ into a corresponding latent state representation $s_t$. The dynamics model processes this latent state alongside an action $a_t$, yielding the subsequent latent state $s_{t+1}$ and an estimated reward $r_t$. Finally, the prediction model accepts a latent state and produces both the predicted policy $p_t$ and the state's value estimate $v_t$. These outputs are instrumental in guiding the agent's action selection process throughout its MCTS. Lastly the agent selects or samples the best action $a_t$ following the searched visit count distribution. During the *training* phase, given a training sequence $\{o_{t-l:t+K}, a_{t+1:t+K}, u_{t+1:t+K}, \pi_{t+1:t+K}, z_{t+1:t+K}\}$ at time $t$ sampled from the replay buffer, where $u_{t+k}$ denotes the actual reward obtained from the environment, $\pi_{t+k}$ represents the target policy obtained through MCTS during the agent-environment interaction, and $z_{t+k}$ is the value target computed using *n-step bootstrapping* Hessel et al. (2018). The representation model initially converts the sequence of observations $o_{t-l:t}$ into the latent state $s_t^0$. Subsequently, the dynamic model executes $K$ latent space rollouts based on the sequence of actions $a_{t+1:t+K}$. The latent state derived after the $k$-th rollout is denoted as $s_t^k$, with the corresponding predicted reward indicated as $r_t^k$. Upon receiving $s_t^k$, the prediction model generates a predicted policy $p_t^k$ and a estimated value $v_t^k$. The final training loss encompasses three components: the policy loss ($l_p$), the value loss ($l_v$), and the reward loss ($l_r$):

$$L_{\text{MuZero}} = \sum_{k=0}^{K} l_p(\pi_{t+k}, p_t^k) + \sum_{k=0}^{K} l_v(z_{t+k}, v_t^k) + \sum_{k=1}^{K} l_r(u_{t+k}, r_t^k) \tag{41}$$

MuZero *Reanalyze*, as introduced in Schrittwieser et al. (2021), is an advanced iteration of the original MuZero algorithm. This variant enhances the model's accuracy by conducting a fresh Monte Carlo Tree Search on sampled states with the most recent version of the model, subsequently utilizing the refined policy from this search to update the policy targets. Such reanalysis yields targets of superior quality compared to those obtained during the initial data collection phase. The Schrittwieser et al. (2021) expands upon this approach, formalizing it as a standalone method for policy refinement. This innovation opens avenues for its application in offline settings, where interactions with the environment are not possible.

## C  IMPLEMENTATION DETAILS

### C.1  ENVIRONMENTS

In this section, we first introduce various types of reinforcement learning environments evaluated in the main paper and their respective characteristics, including different observation/action/reward space and transition functions.

**Atari**: This category includes sub-environments like *Pong, Qbert, Ms.Pacman, Breakout, UpN-Down*, and *Seaquest*. In these environments, agents control game characters and perform tasks based on pixel input, such as hitting bricks in *Breakout*. With their high-dimensional visual input and discrete action space features, Atari environments are widely used to evaluate the capability of reinforcement learning algorithms in handling visual inputs and discrete control problems.

**DMControl**: This continuous control suite comprises 39 continuous control tasks. Our focus here is to validate the effectiveness of ReZero in the continuous action space. Consequently, we have utilized two representative tasks (*ball_in_cup-catch* and *walker-stand*) for illustrative purposes. A comprehensive benchmark for this domain will be included in future versions.

**Board Games**: This types of environment includes *Connect4*, *Gomoku*, where uniquely marked boards and explicitly defined rules for placement, movement, positioning, and attacking are employed to achieve the game's ultimate objective. These environments feature a variable discrete action space, allowing only one player's piece per board position. In practice, algorithms utilize action mask to indicate reasonable actions.

### C.2  ALGORITHM IMPLEMENTATION DETAILS

Our algorithm's implementation is based on the open-source code of LightZero (Niu et al., 2023). Given that our proposed theoretical improvements are applicable to any MCTS-based RL method, we have chosen MuZero and EfficientZero as case studies to investigate the practical improvements in time efficiency achieved by integrating the ReZero boosting techniques: *just-in-time reanalyze* and *speedy reanalyze (temporal information reuse)*.

To ensure an equitable comparison of wall-clock time, all experimental trials were executed on a fixed single worker hardware setting consisting of a single NVIDIA A100 GPU with 30 CPU cores and 120 GiB memory. Besides, we emphasize that to ensure a fair comparison of time efficiency and sample efficiency, the model architecture and hyper-parameters used in the experiments of Section 5 are essentially consistent with the settings in LightZero. For specific hyper-parameters of *ReZero-M* and *MuZero* on Atari, please refer to the Table 3. The main different hyper-parameters in the *DMControl* task are set out in Tables 4. The main different hyper-parameters for the *ReZero-M* algorithm in the *Connect4* and *Gomoku* environment are set out in Tables 5. In addition to employing an LSTM network with a hidden state dimension of 512 to predict the value prefix (Ye et al., 2021), all hyperparameters of ReZero-E are essentially identical to those of ReZero-M in Table 3.

**Wall-time statistics** Note that all our current tests are conducted in the single-worker case. Therefore, the wall-time reported in Table 1 and Table 7 for reaching 100k env steps includes:

- *collect time*: The total time spent by an agent interacting with the environment to gather experience data. Weng et al. (2022) can be integrated to speed up. Besides, this design also makes MCTS-based algorithms compatible to existing RL exploration methods like Burda et al. (2018).

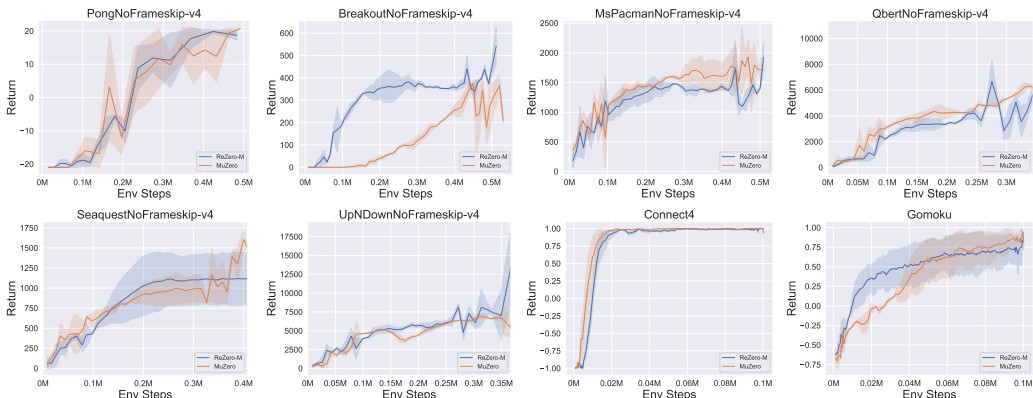

Figure 7: **Sample-efficiency** of ReZero-M vs. MuZero on six representative Atari games and two board games. The horizontal axis represents *Env Steps*, while the vertical axis indicates the *Episode Return* over 5 assessed episodes. ReZero achieves *similar* sample-efficiency than the baseline method. Mean of 5 runs; shaded areas are 95% confidence intervals.

- *reanalyze time*: The time used to reanalyze collected data with the current policy or value function for more accurate learning targets (Schrittwieser et al., 2021).
- *train time*: The duration for performing updates to the agent's policy, value functions and model based on collected data.
- *evaluation time*: The period during which the agent's policy is tested against the environment, separate from training, to assess performance.

Currently, we have set *collect_max_episode_steps* to 10,000 and *eval_max_episode_steps* to 20,000 to mitigate the impact of anomalously long evaluation episodes on time. In the future, we will consider conducting offline evaluations to avoid the influence of evaluation time on our measurement of time efficiency. Furthermore, the ReZero methodology represents a pure algorithmic enhancement, eliminating the need for supplementary computational resources or additional overhead. This approach is versatile, enabling seamless integration with single-worker serial execution environments as well as multi-worker asynchronous frameworks. The exploration of ReZero's extensions and its evaluations in a multi-worker (Mei et al., 2023) paradigm are earmarked for future investigation.

**Board games settings** Given that our primary objective is to test the proposed techniques for improvements in time efficiency, we consider a simplified version of single-player mode in all the board games. This involves setting up a fixed but powerful expert bot and treating this opponent as an integral part of the environment. Exploration of our proposed techniques in the context of learning through self-play training pipeline is reserved for our future work.

# D  ADDITIONAL EXPERIMENTS

## D.1  REZERO-M

In this section, we provide additional experimental results for ReZero-M. As a supplement to Table 1, Table 6 presents the complete experimental results on the 26 Atari environments. Figure 7 displays the performance over environment interaction steps of the ReZero-M algorithm compared with the original MuZero algorithm across six representative Atari environments and two board games. We can find that ReZero-M obtained *similar* sample efficiency than MuZero on the most tasks.

## D.2  REZERO-E

The enhancements of ReZero we have proposed are universally applicable to any MCTS-based reinforcement learning approach theoretically. In this section, we integrate ReZero with EfficientZero to obtain the enhanced ReZero-E algorithm. We present the empirical results comparing ReZero-E with the standard EfficientZero across four Atari environments.

| Hyperparameter | Value |
|---|---|
| Replay Ratio (Schwarzer et al., 2023) | 0.25 |
| Reanalyze frequency | 1 |
| Num of frames stacked | 4 |
| Num of frames skip | 4 |
| Reward clipping (Mnih et al., 2013) | True |
| Optimizer type | Adam |
| Learning rate | $3 \times 10^{-3}$ |
| Discount factor | 0.997 |
| Weight of policy loss | 1 |
| Weight of value loss | 0.25 |
| Weight of reward loss | 1 |
| Weight of policy entropy loss | 0 |
| Weight of SSL (self-supervised learning) loss (Ye et al., 2021) | 2 |
| Batch size | 256 |
| Model update ratio | 0.25 |
| Frequency of target network update | 100 |
| Weight decay | $10^{-4}$ |
| Max gradient norm | 10 |
| Length of game segment | 400 |
| Replay buffer size (in transitions) | 1e6 |
| TD steps | 5 |
| Number of unroll steps | 5 |
| Use augmentation | True |
| Discrete action encoding type | One Hot |
| Normalization type | Layer Normalization |
| Priority exponent coefficient (Schaul et al., 2015) | 0.6 |
| Priority correction coefficient | 0.4 |
| Dirichlet noise alpha | 0.3 |
| Dirichlet noise weight | 0.25 |
| Number of simulations in MCTS (sim) | 50 |
| Categorical distribution in value and reward modeling | True |
| The scale of supports used in categorical distribution (Pohlen et al., 2018) | 300 |

Table 3: Key hyperparameters of **ReZero-M** on *Atari* environments.

| Hyperparameter | Value |
|---|---|
| Replay ratio (Schwarzer et al., 2023) | 0.25 |
| Reanalyze frequency | 1 |
| Batch size | 64 |
| Num of frames stacked | 1 |
| Num of frames skip | 2 |
| Discount factor | 0.997 |
| Length of game segment | 8 |
| Use augmentation | False |
| Number of simulations in MCTS (sim) | 50 |
| Number of sampled actions (Hubert et al., 2021) | 20 |

Table 4: Key hyperparameters of **ReZero-M** on two *DMControl* tasks (*ball_in_cup-catch* and *walker-stand*). More experiments about this hyper-parameter will be explored in the future version. Other unmentioned parameters are the same as that in *Atari* settings.

Figure 8 shows that ReZero-E is better than EfficientZero in terms of time efficiency. Figure 9 indicates that ReZero-E matches EfficientZero's sample efficiency across most tasks. Additionally, Table 7 details training times to 100k environment steps, revealing that ReZero-E is significantly faster than baseline methods on most games.

| Hyperparameter | Value |
|---|---|
| Replay ratio (Schwarzer et al., 2023) | 0.25 |
| Reanalyze frequency | 1 |
| Board size | 6x7; 6x6 |
| Num of frames stacked | 1 |
| Discount factor | 1 |
| Weight of SSL (self-supervised learning) loss | 0 |
| Length of game segment | 18 |
| TD steps | 21; 18 |
| Use augmentation | False |
| Number of simulations in MCTS (sim) | 50 |
| The scale of supports used in categorical distribution | 10 |

Table 5: Key hyperparameters of **ReZero-M** on *Connect4* and *Gomoku* environments. If the parameter settings of these two environments are different, they are separated by a semicolon.

| Env. Name | ReZero-M | MuZero |
|---|---|---|
| Alien | $1.6_{\pm 0.2}$ | $8.6_{\pm 0.4}$ |
| Amidar | $1.5_{\pm 0.2}$ | $8.1_{\pm 0.3}$ |
| Assault | $1.5_{\pm 0.1}$ | $7.5_{\pm 0.1}$ |
| Asterix | $1.3_{\pm 0.1}$ | $7.2_{\pm 0.2}$ |
| BankHeist | $2.9_{\pm 0.3}$ | $8.9_{\pm 0.6}$ |
| BattleZone | $2.2_{\pm 0.3}$ | $9.6_{\pm 0.6}$ |
| ChopperCommand | $3.4_{\pm 0.4}$ | $9.0_{\pm 0.7}$ |
| CrazyClimber | $2.7_{\pm 0.1}$ | $9.1_{\pm 0.4}$ |
| DemonAttack | $1.1_{\pm 0.1}$ | $6.8_{\pm 0.8}$ |
| Freeway | $1.0_{\pm 0.0}$ | $6.1_{\pm 0.2}$ |
| Frostbite | $2.2_{\pm 0.4}$ | $10.9_{\pm 0.8}$ |
| Gopher | $3.2_{\pm 0.6}$ | $8.1_{\pm 0.8}$ |
| Hero | $2.5_{\pm 0.4}$ | $9.9_{\pm 0.6}$ |
| Jamesbond | $2.2_{\pm 0.3}$ | $9.1_{\pm 0.5}$ |
| Kangaroo | $2.0_{\pm 0.2}$ | $8.6_{\pm 0.8}$ |
| Krull | $1.8_{\pm 0.1}$ | $7.7_{\pm 0.3}$ |
| KungFuMaster | $1.3_{\pm 0.1}$ | $7.6_{\pm 0.7}$ |
| PrivateEye | $1.0_{\pm 0.1}$ | $5.8_{\pm 0.5}$ |
| RoadRunner | $1.5_{\pm 0.2}$ | $9.0_{\pm 0.3}$ |
| UpNDown | $1.4_{\pm 0.1}$ | $7.2_{\pm 0.4}$ |
| Pong | $1.0_{\pm 0.1}$ | $4.0_{\pm 0.5}$ |
| MsPacman | $1.4_{\pm 0.2}$ | $6.9_{\pm 0.3}$ |
| Qbert | $1.3_{\pm 0.1}$ | $7.0_{\pm 0.3}$ |
| Seaquest | $1.9_{\pm 0.4}$ | $10.1_{\pm 0.5}$ |
| Boxing | $1.1_{\pm 0.0}$ | $6.6_{\pm 0.1}$ |
| Breakout | $3.0_{\pm 0.8}$ | $4.9_{\pm 1.8}$ |

Table 6: **Average wall-time**(hours) of ReZero-M vs. MuZero on 26 Atari game environments. The time represents the average total wall-clock time to 100k environment steps. Mean and standard deviation over 5 runs.

### D.3 MORE ABLATIONS

This section presents the results of three supplementary ablation experiments. Figure 10 illustrates the impact of using backward-view reanalyze within the ReZero framework on sample efficiency. The results indicate that backward-view reanalyze, by introducing root value as auxiliary information, achieves higher sample efficiency, aligning with the theoretical analysis regarding regret upper bound. Figure 11 demonstrates the effects of different action selection methods during the collect phase. The findings reveal that sampling actions directly from the distribution output by the policy

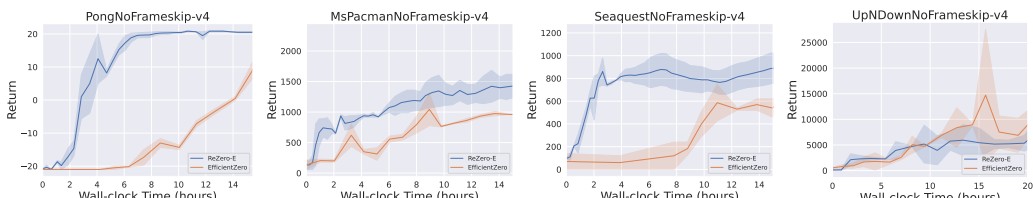

Figure 8: **Time-efficiency** of ReZero-E vs. EfficientZero on four representative Atari games. The horizontal axis represents *Wall-time* (hours), while the vertical axis indicates the *Episode Return* over 5 assessed episodes. ReZero-E achieves higher time-efficiency than the baseline method. Mean of 5 runs; shaded areas are 95% confidence intervals.

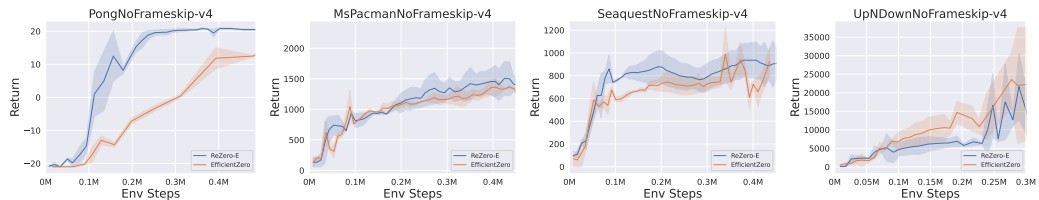

Figure 9: **Sample-efficiency** of ReZero-E vs. EfficientZero on four representative Atari games. The horizontal axis represents *Env Steps*, while the vertical axis indicates the *Episode Return* over 5 assessed episodes. ReZero-E achieves similar sample-efficiency than the baseline method. Mean of 5 runs; shaded areas are 95% confidence intervals.

| avg. wall time (h) to 100k env. steps ↓ | Pong | MsPacman | Seaquest | UpNDown |
|---|---|---|---|---|
| **ReZero-E** (ours) | $\mathbf{2.3}_{\pm 1.4}$ | $\mathbf{3}_{\pm 0.3}$ | $\mathbf{3.1}_{\pm 0.1}$ | $\mathbf{3.6}_{\pm 0.2}$ |
| EfficientZero (Ye et al., 2021) | $10_{\pm 0.2}$ | $12_{\pm 1.3}$ | $15_{\pm 2.3}$ | $15_{\pm 0.7}$ |

Table 7: **Average wall-time** of ReZero-E vs. EfficientZero on four Atari games. The time represents the average total wall-time to 100k environment steps for each algorithm. Mean and standard deviation over 5 runs.

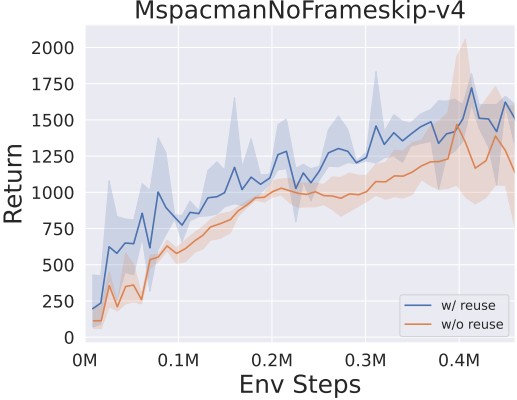

Figure 10: The results of the ablation study comparing the use of backward-view reanalyze versus its absence. Mean of 3 runs; shaded areas are 95% confidence intervals.

network does not significantly degrade the experimental results compared to using MCTS for action selection. Figure 12 depicts the relationship between the average MCTS search duration and the batch size. We set a baseline batch size of 256 and experimented with search sizes ranging from 1 to 20 times the baseline, calculating the average time required to search 256 samples by dividing the total search time by the multiplier. The results suggest that larger batch sizes can better leverage the advantages of parallelized model inference and data processing. However, when the batch size becomes excessively large, constrained by the limits of hardware resources (memory, CPU), the search

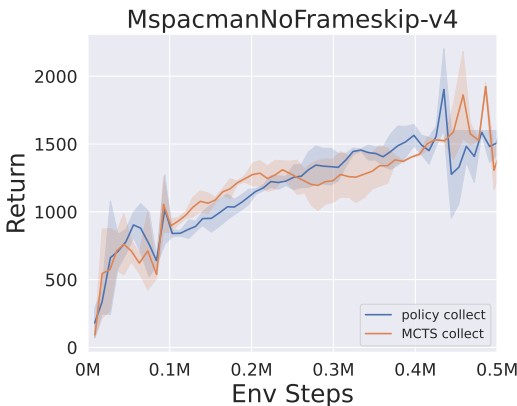

Figure 11: The comparison of the outcomes of sampling actions based on the policy network's output against selecting actions using MCTS. Mean of 3 runs; shaded areas are 95% confidence intervals.

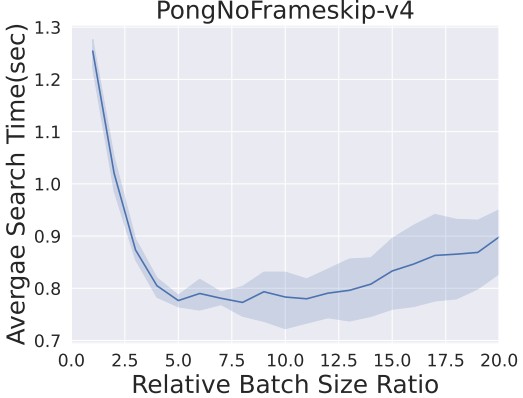

Figure 12: The depiction of the variation in the average search speed of MCTS as the batch size increases. Mean of 3 runs; shaded areas are 95% confidence intervals.

speed cannot increase further and may even slightly decrease. We ultimately set the batch size to 2000, which yields the fastest average search speed on our device.

### D.4 TOY CASE

We offer the complete code for the toy case. Readers can compare the core functions $search()$ and $reuse\_search()$ of the MCTS class to understand how root node values are utilized and compare $select()$ and $reuse\_select()$ to understand how the search process is prematurely halted.

```python
import random
import math
import time
import numpy as np
import matplotlib.pyplot as plt

class Node:
    def __init__(self, state, parent=None):
        self.state = state
        self.parent = parent
        self.children = []
        self.visits = 0
        self.value = 0

    def is_fully_expanded(self):
        return len(self.children) == len(self.state.get_possible_actions())

    def add_child(self, child_state):
        child = Node(child_state, self)
        self.children.append(child)
        return child

class MCTS:
    def __init__(self, exploration_weight=1.0):
        self.exploration_weight = exploration_weight
        self.gamma = 0.9

    # the search process in origin MCTS
    def search(self, initial_state, max_iter=100):
        root = Node(initial_state)

        for _ in range(max_iter):
            node = self.select(root)
            reward = self.simulate(node.state)
            self.backpropagate(node, reward)

        best_child = self.get_best_child(root, 0)
        return best_child.state.current_pos, root.value

    # the search process in our accelerated MCTS by reuse the root value
    def reuse_search(self, initial_state, value, action, max_iter=100):
        root = Node(initial_state)

        for _ in range(max_iter):
            node = self.reuse_select(root,action)
            if node.state.current_pos == action:
                reward = value
            else: reward = self.simulate(node.state)
            self.backpropagate(node, reward)
        best_child = self.get_best_child(root, 0)
        return best_child.state.current_pos, root.value

    # select nodes in origin MCTS
    def select(self, node):
        while not node.state.is_terminal():
            if not node.is_fully_expanded():
                return self.expand(node)
            else:
                node = self.get_best_child(node, self.exploration_weight)
        return node

    # select nodes in our accelerated MCTS by reuse the root value
    def reuse_select(self, node,actionpos):
        while not (node.state.is_terminal() or node.state.current_pos == actionpos):
            if not node.is_fully_expanded():
                return self.expand(node)
            else:
                node = self.get_best_child(node, self.exploration_weight)
        return node

    def expand(self, node):
        actions = node.state.get_possible_actions()
        for action in actions:
            if not any(child.state.current_pos == node.state.copy().step(action)[0] for child in node.children):
                new_state = node.state.copy()
                new_state.step(action)
                return node.add_child(new_state)
        return None

    def simulate(self, state):
        sim_state = state.copy()
```

```
1242        count = 1
1243        while not sim_state.is_terminal():
1244            action = random.choice(sim_state.get_possible_actions())
             sim_state.step(action)
1245            count += 1
1246        return sim_state.get_reward()/count

1247    def backpropagate(self, node, reward):
         gamma = self.gamma
1248        while node is not None:
            node.visits += 1
1249            node.value += reward * gamma
            node = node.parent
1250            gamma *= self.gamma
1251
1252    def get_best_child(self, node, exploration_weight):
         best_value = -float('inf')
1253        best_children = []
         for child in node.children:
1254            exploit = child.value / child.visits
            explore = math.sqrt(2.0 * math.log(node.visits) / child.visits)
1255            value = exploit + exploration_weight * explore
            if value > best_value:
1256                best_value = value
                best_children = [child]
1257            elif value == best_value:
                best_children.append(child)
1258        return random.choice(best_children)
1259
1260 class GridWorld:
     def __init__(self):
1261        self.grid = [[0 for _ in range(4)] for _ in range(4)]
         self.start_pos = (0, 0)
1262        self.goal_pos = (0, 3)
         self.current_pos = self.start_pos
1263        self.actions = ['down', 'up', 'left', 'right']
1264
1265    def reset(self):
         self.current_pos = self.start_pos
1266        return self.current_pos
1267    def step(self, action):
         if action not in self.actions:
1268            raise ValueError("Invalid_action")
1269
         x, y = self.current_pos
1270
         if action == 'up':
1271            x = max(0, x - 1)
         elif action == 'down':
1272            x = min(3, x + 1)
         elif action == 'left':
1273            y = max(0, y - 1)
         elif action == 'right':
1274            y = min(3, y + 1)
1275
         self.current_pos = (x, y)
1276        reward = 1 if self.current_pos == self.goal_pos else 0
         done = self.current_pos == self.goal_pos
1277
1278        return self.current_pos, reward, done
1279
1280    def is_terminal(self):
         return self.current_pos == self.goal_pos
1281
1282    def get_reward(self):
         return 1 if self.current_pos == self.goal_pos else 0
1283
     def get_possible_actions(self):
1284        x, y = self.current_pos
         possible_actions = []
1285
         if x > 0:
1286            possible_actions.append('up')
         if x < 3:
1287            possible_actions.append('down')
         if y > 0:
1288            possible_actions.append('left')
         if y < 3:
1289            possible_actions.append('right')
1290
         return possible_actions
1291
1292    def copy(self):
         new_grid = GridWorld()
1293        new_grid.current_pos = self.current_pos
         return new_grid
1294
1295    def render(self):
```

```python
        for i in range(4):
            for j in range(4):
                if (i, j) == self.current_pos:
                    print("A", end=" ")
                elif (i, j) == self.goal_pos:
                    print("G", end=" ")
                else:
                    print(".", end=" ")
            print()
        print()

class GridWorldWithWalls:
    def __init__(self):
        self.grid = [[0 for _ in range(7)] for _ in range(7)]
        self.start_pos = (0, 0)
        self.goal_pos = (6, 6)
        self.current_pos = self.start_pos
        self.actions = ['down', 'up', 'left', 'right']
        self.walls = [(2, 2), (2, 3), (1,1), (3, 2), (3, 4), (3,2), (0,5), (4, 4), (6,3), (5,3)]

        for wall in self.walls:
            self.grid[wall[0]][wall[1]] = 1

    def reset(self):
        self.current_pos = self.start_pos
        return self.current_pos

    def step(self, action):
        if action not in self.actions:
            raise ValueError("Invalid action")

        x, y = self.current_pos

        if action == 'up':
            x = max(0, x - 1)
        elif action == 'down':
            x = min(6, x + 1)
        elif action == 'left':
            y = max(0, y - 1)
        elif action == 'right':
            y = min(6, y + 1)

        if (x, y) not in self.walls:
            self.current_pos = (x, y)

        reward = 1 if self.current_pos == self.goal_pos else 0
        done = self.current_pos == self.goal_pos

        return self.current_pos, reward, done

    def is_terminal(self):
        return self.current_pos == self.goal_pos

    def get_reward(self):
        return 1 if self.current_pos == self.goal_pos else 0

    def get_possible_actions(self):
        x, y = self.current_pos
        possible_actions = []

        if x > 0 and (x - 1, y) not in self.walls:
            possible_actions.append('up')
        if x < 6 and (x + 1, y) not in self.walls:
            possible_actions.append('down')
        if y > 0 and (x, y - 1) not in self.walls:
            possible_actions.append('left')
        if y < 6 and (x, y + 1) not in self.walls:
            possible_actions.append('right')

        return possible_actions

    def copy(self):
        new_grid = GridWorldWithWalls()
        new_grid.current_pos = self.current_pos
        return new_grid

    def render(self):
        for i in range(7):
            for j in range(7):
                if (i, j) == self.current_pos:
                    print("A", end=" ")
                elif (i, j) == self.goal_pos:
                    print("G", end=" ")
                elif (i, j) in self.walls:
                    print("#", end=" ")
                else:
                    print(".", end=" ")
            print()
        print()
```

```
1350
1351
1352     env = GridWorldWithWalls()
         mcts = MCTS()
1353
         # plot the grid
1354     plt.figure(figsize=(6, 6))
1355     plt.matshow(env.grid, cmap='binary', fignum=0, vmin=0, vmax=1)

1356     for i in range(7):
             for j in range(7):
1357             if (i, j) == env.start_pos:
1358                 plt.text(j, i, 'A', ha='center', va='center', color='black', fontsize=12)
                 elif (i, j) == env.goal_pos:
1359                 plt.text(j, i, 'G', ha='center', va='center', color='black', fontsize=12)
                 elif (i, j) in env.walls:
1360                 plt.text(j, i, '', ha='center', va='center', color='black', fontsize=12)
                 else:
1361                 plt.text(j, i, '', ha='center', va='center', color='black', fontsize=12)
1362
         for wall in env.walls:
1363         plt.gca().add_patch(plt.Rectangle((wall[1] - 0.5, wall[0] - 0.5), 1, 1, color='black'))
1364
         for i in range(7):
1365         for j in range(7):
                 plt.gca().add_patch(plt.Rectangle((j - 0.5, i - 0.5), 1, 1, fill=False, edgecolor='black', linewidth=2))
1366
         plt.title('GridWorld_Environment', fontsize=24)
1367     plt.xticks(np.arange(7), ['0', '1', '2', '3', '4', '5', '6'])
         plt.yticks(np.arange(7), ['0', '1', '2', '3', '4', '5', '6'])
1368     plt.xlabel('Column', fontsize=24)
         plt.ylabel('Row', fontsize=24)
1369     plt.show()
1370
         # record the search time
1371     search_times = np.zeros((7, 7))
1372     reuse_times = np.zeros((7, 7))
         for i in range(7):
1373         for j in range(7):
1374             if (i, j) == env.goal_pos or (i,j) in env.walls:
                     continue
1375
                 env.current_pos = (i, j)
1376             print(f"Starting_MCTS_from_position:_{env.current_pos}")
1377
                 start_time = time.time()
1378             reuse_action, root_value = mcts.search(env)
                 end_time = time.time()
1379             search_time = end_time - start_time
1380             search_times[i, j] = search_time

1381             env.current_pos = reuse_action
                 if reuse_action == env.goal_pos:
1382                 reuse_value = 1
                 else: _, reuse_value = mcts.search(env)
1383
                 env.current_pos = (i, j)
1384             start_time = time.time()
1385             best_action, root_value = mcts.reuse_search(env, reuse_value, reuse_action)
                 end_time = time.time()
1386             reuse_time = end_time - start_time
1387             reuse_times[i, j] = reuse_time
1388
1389
                 print(f"Best_action_leads_to_position:_{reuse_action}")
1390             print(f"Reuse_search_best_action_leads_to_position:_{best_action}")
                 print(f"Search_time:_{search_time:.4f}_seconds")
1391             print(f"reuseSearch_time:_{reuse_time:.4f}_seconds\n")
1392
         # plot the time heatmap
1393     plt.figure(figsize=(6, 6))
         plt.imshow(search_times, cmap='viridis', vmin=0, vmax=0.2, interpolation='nearest')
1394     plt.colorbar(label='Search_Time_(seconds)')
         plt.title('Origin_Search_Time', fontsize=20)
1395     plt.xticks(np.arange(7), ['0', '1', '2', '3', '4', '5', '6'])
1396     plt.yticks(np.arange(7), ['0', '1', '2', '3', '4', '5', '6'])
         plt.xlabel('Column', fontsize=20)
1397     plt.ylabel('Row', fontsize=20)
         plt.figure(figsize=(6, 6))
1398     plt.imshow(reuse_times, cmap='viridis', vmin=0, vmax=0.2, interpolation='nearest')
1399     plt.colorbar(label='Search_Time_(seconds)')
         plt.title('Accelerated_Search_Time', fontsize=20)
1400     plt.xticks(np.arange(7), ['0', '1', '2', '3', '4', '5', '6'])
1401     plt.yticks(np.arange(7), ['0', '1', '2', '3', '4', '5', '6'])
         plt.xlabel('Column', fontsize=20)
1402     plt.ylabel('Row', fontsize=20)
         plt.show()
1403
```

