# OpenReview forum: "ReZero: Boosting MCTS-based Algorithms by  Backward-view and Entire-buffer Reanalyze"
_ICLR.cc/2025/Conference — Submitted to ICLR 2025_

### Official Review · Reviewer_MkuL · 2024-10-25

**Soundness:** 2
**Presentation:** 2
**Contribution:** 1
**Rating:** 5
**Confidence:** 5

**Summary:**

This research presents an innovative approach to enhance the efficiency of Monte Carlo Tree Search (MCTS) algorithms, like MuZero, by introducing a method called ReZero. By leveraging a backward-view reuse technique and periodically reanalyzing the entire buffer, the authors aim to reduce search costs while maintaining or even improving performance. This work promises to simplify data collection and reanalysis in various decision-making domains.

**Strengths:**

This work proposes the backward-view reuse technique, which reutilizes previously collected data to boost search efficiency. Additionally, the entire-buffer reanalyze mechanism represents a creative adaptation of traditional reanalyze processes, allowing for more effective data utilization. This originality is further underscored by the application of these techniques across diverse environments.The paper is well-structured and clearly written.

**Weaknesses:**

The main weakness of the paper lie in the lack of innovation in the method and the scarcity of baseline comparisons in the experiment. The reuse method has some other approaches in previous MCTS works, but this paper fails to conduct a sufficient disscusion. Additionally, the experiments in the article only compare with the MuZero algorithm. Whether the EfficientZero or Sampled MuZero would show improvements when using this method requires further experimental illustration.

**Questions:**

Main Questions:
1) Previous studies such as "Information capture and reuse strategies in Monte Carlo Tree Search, with applications to games of hidden information" and "Learning policies from self-play with policy gradients and MCTS value estimates" have also proposed methods similar to reuse in MCTS. What are the differences between the methods in this work and those in the previous studies? This requires detailed discussion.
2) The paper presents some theoretical analyses and proofs, but I fail to understand the specific connection between this proof and the method proposed in the article, and a detailed explanation is needed.
3) How the proposed method performs in terms of improvement on Sampled MuZero and EfficientZero requires further experimental illustration.

Some minor issues:
1) It is recommended that the algorithm not use the pseudo-code of Python in Algorithm 1.
2) Line 235: what does $S^{0}s$ refer to?
3) The font size of the text in Figure 3 is too small.
4) Is it understandable that the experimental effect of Figure 6 is not obvious and there is only a small amount of influence?
5) Why is the decision made to periodically reanalyze the entire buffer instead of frequently reanalyzing small batches of data? How does this approach impact search efficiency?

---

> ### Author Response · Authors · 2024-11-20
> **Regarding the Baseline**
>
> Dear Reviewer, thank you for your feedback! In the paper, we used MuZero, EfficientZero, and Sampled MuZero as examples to verify the time efficiency improvements of ReZero. The results in Figure 5 over two continuous DMC control tasks are from Sampled MuZero, but due to our oversight, the legend was not correctly labeled. This will be corrected in the revised version of the paper. Additionally, the improvements of ReZero relative to EfficientZero are presented in Appendix D. These experimental results demonstrate the potential of ReZero as a general method for improving the time efficiency of MCTS+RL algorithms.

---

> ### Author Response · Authors · 2024-11-20
> **Regarding the differences between ReZero and those in the previous studies**
>
> As we mentioned in Section 2.2 of original paper, we have compared our approach with several existing MCTS acceleration techniques, such as the state abstraction method in PTSAZero [1] and the simple reuse technique in KataGo [2]. It is important to clarify that we do not claim our backward-view reanalyze strategy is superior to these methods. Instead, we propose a novel perspective on information reuse from a backward-view standpoint.
>
> Regarding the two papers mentioned by the reviewer, the former aims to formalises the correlation between states and actions, and empirically investigate combination operators for MCTS enhancements for the correlate states in analogous positions within different episodes. This work is quite similar to our comparad work PTSAZero, our backward-view analyze can build on the top of their correlate states or state abstractions. The latter "Learning policies from self-play with policy gradients and MCTS value estimates" mainly derives a policy gradient expression with MCTS value estimates rather than MCTS visit counts. The reuse technique they use is just a naive version to reuse search trees from previous turns (similar to KatoGo). Therefore, we believe that our approach complements rather than competes with these methods, offering a complementary angle that can enhance the overall efficiency of MCTS.
>
> [1] Fu, Yangqing, et al. "Accelerating monte carlo tree search with probability tree state abstraction." Advances in Neural Information Processing Systems 36 (2023): 61008-61019.
>
> [2] Wu, David J. "Accelerating self-play learning in go." arXiv preprint arXiv:1902.10565 (2019).

---

> ### Author Response · Authors · 2024-11-20
> **Regarding the "specific connection between this proof and the method proposed in the article"**
>
> Regarding the relationship between our theoretical analyses and the practical algorithm, we have already detailed this connection in Section 4 of our original paper. Here, we aim to elucidate this relationship in a more accessible manner.
>
> Our backward-view reanalyze is grounded in a straightforward insight: if we could know the true state-value (e.g., expected long-term return) of a child node, we could potentially bypass the need for further search, thereby conserving computational resources and concentrate the search budget into more valuable part. In Section 4.1, we first provide a comprehensive description of the entire algorithm, aiming to present a complete overview of the practical implementation. However, as stated at the outset of Section 4.2, our proposed algorithm introduces distinctions from the classic MCTS process employed in AlphaZero. Consequently, it is not reliable to directly apply and scale our algorithm to concrete RL problems without further validation.
>
> To address this, we have conducted theoretical analyses and proofs to substantiate the reliability and efficacy of our backward-view reanalyze. Specifically, we have demonstrated that, as the total number of visits increases, our backward-view design progressively concentrates the visit distribution on the optimal arm of a non-stationary bandit (Theorem 1). This conclusion is further extended to the settings of AlphaZero in a similar manner (Appendix A). These theoretical analyses alleviate concerns that our method might be constrained by local optima or low-quality data, providing evidence for algorithm convergence and potential performance enhancements.
>
> It is important to note that the theoretical analyses and algorithm design were not conducted in a strictly sequential order. Instead, they were developed in a complementary and iterative process, with each influencing the other. However, for the sake of clarity and practical application, we chose to first outline the algorithm's workflow in the paper and subsequently analyze its theoretical reliability. If the above explanation satisfactorily addresses the reviewer's concerns, we are prepared to revise our paper based on these newly added explanations to further enhance the presentation of this section.

---

> ### Author Response · Authors · 2024-11-20
> **Regarding Figure 6**
>
> Regarding Figure 6, due to space constraints, our explanation in the paper focuses primarily on the core conclusion: the appropriate reanalyze frequency can save time overhead without causing any noticeable performance loss. This is evident from our ReZero's performance being close to that of the reanalyze ratio=1 scenario. However, we acknowledge that the detailed analysis and explanation of the experimental groups were indeed brief. We have addressed this issue and provided a more comprehensive explanation here (with relevant content also added to the paper).
>
> Specifically, as shown in the right panel of Figure 6, with the x-axis representing Env Steps, MuZero reanalyze ratio=1 exhibits a significant performance improvement over reanalyze ratio=0 (comparing the purple line to the blue line). For our ReZero method, increasing the frequency from 0 to 1/3 to 1 progressively enhances performance (comparing the blue line to the orange line to the green line), with the frequency of 1 nearly matching the performance upper bound represented by the purple line. Further increasing the frequency from 1 to 2 (comparing the green line to the red line) shows no significant performance improvement, indicating that our algorithm's buffer reanalyze frequency needs only to reach a moderate level to be effective. A frequency of 1 is thus a recommended value, eliminating the need for additional hyperparameter tuning. On the other hand, the left panel of Figure 6 validates the effect when the x-axis represents wall-clock time. A suitable frequency (such as 1) significantly improves episode return within the same time frame compared to the purple, blue, and orange lines. We hope this detailed explanation clarifies the experimental results and addresses the reviewer's concerns.

---

> ### Author Response · Authors · 2024-11-20
> **Regarding "Why is the decision made  to periodically reanalyze the entire buffer instead of frequently reanalyzing small batches of data"**
>
> If one were to opt for high-frequency reanalyze of small batch data, this would introduce additional hyperparameters that require careful tuning: the size of the batch to be reanalyzed, the selection criteria for this batch, and the frequency of reanalyze. This not only increases the number of hyperparameters but also couples the reanalyze process with the sampling strategy for training, such as priority sampling. In contrast, our periodic reanalyze requires control over only a single hyperparameter: the frequency.
>
> Moreover, we can draw parallels to similar ideas in other RL studies. Reanalyze fundamentally involves using the latest network to obtain better training targets. This concept is analogous to the recompute Advantage operation in modern PPO implementations, where the entire collected data is processed together. Similarly, the target network in DQN is updated as a whole after a certain number of iterations, with the latest weights replacing the old ones. There is rarely a need to design specialized algorithms for small batch data, likely due to the high variance associated with small batches in online RL, making it challenging to develop stable algorithmic techniques. We believe this explanation clarifies the rationale behind our approach and addresses the reviewer's concerns regarding the complexity and stability of high-frequency reanalyze of small batch data.

---

> ### Author Response · Authors · 2024-11-20
> **Regarding the Explanation of \(S^0\)**
>
> As shown in the right diagram of Figure 2, we sample \(n + 1\) trajectories of length \(k + 1\), forming a batch and performing a search in the reverse direction of the trajectories. Therefore, $S^0$s refers to the first-step state of all sampled trajectories, which is also the state contained in the first row of the right diagram in Figure 2.

---

> ### Author Response · Authors · 2024-11-20
>
> We are very grateful for your valuable feedback and hope this addresses your concerns. If you have any further suggestions, we would be very grateful for your feedback. Thank you very much.

---

### Official Review · Reviewer_bENq · 2024-11-03

**Soundness:** 1
**Presentation:** 2
**Contribution:** 2
**Rating:** 3
**Confidence:** 4

**Summary:**

Algorithms like MuZero extend the use of MCTS to environments without known models. However, its extensive tree search incurs a substantial time overhead. This pressing matter has motivated many research papers to propose mechanisms mitigating this wall-clock time problem. The authors in this paper propose a new algorithm of this sort, called ReZero, which is orthogonal to the previous contributions, thus making it readily deployable with many of the previous works. ReZero leverages a backward-view reanalyse technique that prunes the exploration of certain specific nodes by using previously searched root values. ReZero also reanalyses the whole buffer periodically after a fixed number of training iterations. In tandem with this reanalyse technique, ReZero employs a search strategy that is akin to a one-armed bandit algorithm. The authors then analyse ReZero from a theoretical point of view. In Theorem 1, a bound on the expected number of suboptimal action visits is proposed. Moreover, this bound implies that this expected number of suboptimal action visits is sublinear, i.e. $\lim_{n \rightarrow \infty}\frac{\mathbb{E}\left[ T_i(n)\right]}{n} = 0$. In the Appendix, the authors also claim to provide such bound for AlphaZero. Empirical results seem to indicate that ReZero-M, which is ReZero employed with MuZero with SSL, can significantly decrease the wall-clock time compared to MuZero. Similar empirical results are stated in the Appendix comparing ReZero-E with EfficientZero.

**Strengths:**

- The subject that the paper addresses is very important. Indeed, improving the sample efficiency of MCTS methods such as MuZero, along with the wall-clock time they incur during tree search could dramatically improve the current MCTS methods.
- Explaining MCTS methods can be a difficult task, thus using illustrations such as Figure 2 helps a lot in explaining the approach.
- There is an attempt at providing finite-time theoretical analysis, in the form of a bound on the expected number of visits to suboptimal actions. Such attempts are common within the Multi-Armed Bandits literature but are less frequent in MDP contexts. It is therefore appreciated that the authors made this theoretical effort.
- The experiments, if they are reproducible with a publicly available code, seem to support the authors claims. However, I am not confident enough to assess the experimental setup accurately.
- The contribution is orthogonal to previously proposed solutions. This implies that ReZero is readily applicable with the other proposed methods from the literature.

**Weaknesses:**

I see two significant weaknesses with this paper. Lack of clarity and mistakes in the theoretical result.

## Lack of Clarity:

- In Algorithm 1 there are many undefined functions: prepare, origin\_MCTS, select\_root\_child, traverse . I did not find these defined in the Appendix either. It is hard for the reader to understand the algorithmic steps from the provided code. I suggest that the authors rather provide an abstract pseudocode or thorough explanations of each step in an iteration, accompanied by some simple illustration. I struggle to deeply understand ReZero, even now I am unsure that I grasp it sufficiently.
- From Figure 2, the considered MDPs seem to have deterministic transition dynamics. This is also implied by Eq. (5). Is ReZero specific to deterministic transition dynamics? If that is the case, then it should be mentioned while introducing the method. Otherwise, Eq. (5) should account for the stochasticity of transitioning to a certain state through a (state, action) pair.
- Section 4.1 is hard to follow. Does ReZero follow the standard MCTS steps: Selection, Expansion, Simulation and Backpropagation? How is the data collected in the replay buffer? In Section 4.3, a policy network will be briefly mentioned, which implies the use of DeepRL. It makes hard for the reader to follow the explanation with scattered information like this. I only realised that there is a DeepRL approach later on, which confused me. What is the network approximating? Is it just the policy or the value function as well? How are the targets constructed in this case? And how are the values updated? There are too many questions here related to the algorithmic machinery itself that I think the authors should spend time explaining. As a suggestion, I think that the authors should delete section 3.1, which I did not find helpful, and spend more space and effort on clearly stating and explaining the algorithmic steps. Figures like Figure 1 and 2 could also be very helpful here, even if put in an illustrative appendix. Figure 3 to me does not seem standalone, it could be useful but only accompanied by this large section just defining the algorithm fully.
- It is unclear how the mini batches come to existence. Do they stem from filling the replay buffer by following the Tree and Simulation policies (as per the standard MCTS terminology)? If so, how are all the trajectories in Figure 2 of the same length? Is this just for illustrative purposes or are they indeed of the same length?
- Could you provide a more thorough explanation of traverse in Line 240? Is it following the Tree and Simulation policies? When does traverse happen compared to reanalyse? I thought it happened at the end of reanalyse but Algorithm 1 suggests otherwise.
- What if during reanalyse, two actions (or more) $a_1, a_2$ have been taken at state $S_l^t$ (in different trajectories)? Would we have $I_l^t\left( a_1\right) = r_1 + \gamma m_1, I_l^t\left( a_2\right) = r_2 + \gamma m_2$? in which case, the policy in (4) and (5) will not taken into consideration the UCB scores of these actions but rather their estimates directly. If many actions of this sort are explored, then how would exploration occur? I think I am missing a crucial detail about the algorithm here, hence why I suggest that the authors restructure the paper in a way that prioritises much more a clear explanation of ReZero.
- In (5), $I_t\left( a\right)$ should be denoted $I_t^l\left( a\right)$ for consistency of the notation with (4).
- This is not a clarity issue per se, nevertheless it should be mentioned. Regarding the non-stationary bandit in (2), the concentration assumption should be around $\mu_{is} = \mathbb{E}\left[ \overline{X_{is}} \right]$ rather than $\mu_i$ for a meaningful analysis. It is true that the text in the paper "Bandit-based Monte Carlo Planning" is unclear about this. In page 5, the authors just "assume the tail inequalities (3), (4)" and we it is unclear whether they mean $\mu_{is}$ or $\mu_i$. However, in a follow-up journal publication by the same authors "Improved Monte Carlo Methods" where they thoroughly rewrite the previous paper and correct some of its theoretical claims, the authors clarify this misunderstanding in Assumption 1. They indeed consider the concentration to be around $\mu_{is}$. If $\delta_{is} = \mu_{is} - \mu_i = \mathcal{o}\left( \sqrt{\frac{1}{s}}\right)$ the Assumption 1 would indeed imply (2) in the reviewed paper for an appropriate constant $C$. However, I believe that no such assumption was made.

To recapitulate this section. I believe that most of the confusion comes from a lack of investment in clearly explaining the algorithm ReZero. I suggest that the authors take this into consideration when revising the paper. Some of the ways to make this improvement is through illustrations, pseudocode and accompanying text. Some sections like 3.1 could be deleted in favour of this restructuring.

## Theoretical mistakes:

I love the fact that the authors committed time to theoretically analyse their algorithm, I encourage such endeavour. I would like to draw the attention of the authors to a number of mistakes in their proofs that unfortunately end up not supporting their claims. At the end of this section, I will provide references that I hope will be helpful to the authors in rethinking their proof approach in the future.
- First, the statement of Theorem 1 is not rigorous. The terms $P_i$ are nowhere defined. I understood from context and the proof that they are the prior terms $P\left( s, a\right)$ introduced in $3$, nevertheless, they should be properly defined in the theorem statement. The same remark applies to other terms like $\widehat{\mu}_{ln}$. Moreover, and even more importantly, What does it mean precisely to choose actions according to $(5)$? A first understanding would lead the reader to believe that once an arm $i$ is played, its index $I_i^t$ will always be equal to its reward, thus removing the UCB exploration bonus, but it seems to me unlikely that this is what is meant by the authors as it would imply full exploitation algorithm. Please provide the precise statement of how actions are chosen.
- General advice that the authors could incorporate straightforwardly. Please start using \left( , \right) , \Bigg\{ , \Bigg\} ... for clearer mathematical notation.

### Lemma 1:
- Minor: Line 706, put a space between "Let" and "$\widehat{\mu}_t$".
- Minor: I think the term $\exp\left( \frac{\sqrt{a}\epsilon}{C^2}\right)$ should rather be $\exp\left( 2\frac{\sqrt{a}\epsilon}{C^2}\right)$ in (13). This is a minor detail as it does not change the result, but could you please verify this?

### Theorem 1:
- $\widehat{\mu_{is}}$ is the average of the first $s$ samples of arm $i$, thus I think that the notation $\widehat{\mu}_{ln}$, in the theorem statement, is inadequate as it implies that arm $l$ has been chosen $n$ times. It should rather be replaced by $\widehat{\mu}_l\left( n\right)$.
- $T_i\left( n\right) = \sum_{t=1}^n \mathbb{I}\\{ A_t = i \\}$, thus $A_t$ denotes the chosen arm at time $t$. This definition should be stated during the proof.
-  I believe there is a mistake in (24). For the passage from (23) to (24), the authors employ $\sum_{t=2}^{n}\exp\left( -a\sqrt{t-1}\right) \le \int_{t=1}^\infty \exp\left( -a\sqrt{t-1}\right)dt$ but they do not make the calculation explicitly. I do it below via change of variable $u = \sqrt{t-1}$:
\begin{align*}
\int_{t=1}^\infty \exp\left( -a\sqrt{t-1}\right)dt &= \int_{u=0}^\infty 2u\exp\left( -au\right)du\\\\
&= \left[ -\frac{2u}{a}\exp\left( -au\right)\right]_{0}^\infty + \int_0^\infty \frac{2}{a}\exp\left( -au\right)du\\\\
&= \frac{2}{a}\left[ -\frac{1}{a}\exp\left( -au\right)\right]_0^\infty = \frac{2}{a^2}\\\\
\implies \sum_2^n\exp\left( -\frac{1}{C^2}\epsilon P_1 \sqrt\{t-1}\right) &\le \frac{2C^4}{\epsilon^2 P_1^2}
\end{align*}
This in turn implies that the term in (24) should be replaced with $1 + \frac{2C^6}{\epsilon^4 P_1^2}$.
- The passage from (26) to (27) is wrong, and this is what will unfortunately go against your claim. I invite the authors to discuss this important specific point with me during the rebuttal. There is an omitted sum here. When upper bounding the term in (26), you should proceed as follows:
$$
\mathbb{E}\left[ \sum_{t=1}^n\sum_{s=1}^{t-1} \mathbb{I}\Bigg\\{ \widehat{\mu}_{is} + P_i\sqrt{\frac{n-1}{\left( 1 + s\right)^2}} \ge \mu_1 - \epsilon\Bigg\\}, T_i\left( t-1\right) = s\right]
$$
Then this term will be further upper bounded by omitting the event $T_i\left( t-1\right) = s$. For a similar illustrative example of this sum, please refer to the paper "Finite-time Analysis of the Multiarmed Bandit Problem", especially the proof of Theorem 1, check (6). Now this double sum will lead to the following:
$$
(25) \le n + \frac{2\left( n-1\right)\sqrt{P_i^2 \left( n-1\right)}}{\Delta_i - \epsilon} + \frac{\left( n-1\right)C^2}{\left( \Delta_i - \epsilon\right)^2}\exp\left( \frac{2\left( \Delta_i -  \epsilon\right)\sqrt{P_i^2 \left( n-1\right)}}{C^2}\right)
$$
Thus leading to:
$$
\mathbb{E}\left[ T_i\left( n\right)\right] \le 1 + \frac{2C^6}{\epsilon^4 P_1^2} + n + \frac{2\left( n-1\right)\sqrt{P_i^2 \left( n-1\right)}}{\Delta_i - \epsilon} + \frac{\left( n-1\right)C^2}{\left( \Delta_i - \epsilon\right)^2}
$$
Now unfortunately the term $n + \frac{2\left( n-1\right)\sqrt{P_i^2 \left( n-1\right)}}{\Delta_i - \epsilon} + \frac{\left( n-1\right)C^2}{\left( \Delta_i - \epsilon\right)^2}$ is not sublinear and as such you can no longer deduce that $\mathbb{E}\left[ T_i\left( n\right)\right]$ is sublinear. In fact that is the reason why UCB, in the Multi-Armed Bandit (MAB) setting, employs a logarithmic term in the index $\widehat{\mu}_i\left( t-1\right) + C\sqrt{\frac{\log t}{1 + T_i\left( t-1\right)}}$ as opposed to a polynomial term. $\log t$ slows down exploration thus leading to the concentration of visits to the optimal arm. With a polynomial term in $t$, this is no longer guaranteed because the index grows quickly leading to too much exploration even for the suboptimal arms. Now the reason AlphaZero employs a polynomial term instead of a logarithmic term is because there is a need for a lot more exploration in MDPs than it is for MABs. You need a lot more samples to estimate the value of a node in an MDP than you need to estimate the value of an arm in a MAB.
- Suggestions to solve this issue. I think there might be a need to change the way you define the index of your UCB. For finite-sample analyses of UCT-like algorithms, please check the following paper "Nonasymptotic Analysis of Monte Carlo Tree Search".

## Experiments:

I do not know how to accurately assess the experimental setup as I feel like I'm not fully grasping the algorithm itself to pinpoint the merits of its contributions in the experiments. Nevertheless, if ReZero-M is just ReZero applied to MuZero, then it does hint at some substancial speedups. My concerne is with variance. From my understanding of your reanalyse strategy, you can stop searching a node prematurely, and with less samples to estimate its values, the variance of this estimate could be important. I think the authors should spend some space discussing this issue. Figure 7, SeaquestNoFrameskip-v4, Gomoku do hint at this phenomenon, while we see that MuZero's variance is stable in these cases.

## Misspelling:

There are a multitude of misspellings in the paper. Although this is a minor issue, the authors should take the time to revise their paper accordingly.
- Line 308: "in in".
- Line 059: "we aims".
- Line 084: "a efficient"
- Line 127: "a last observation sequences"
- Line 137: "we suggests"
- Line 181: "reward $r_A$", shouldn't it be "return $r_A$"?
- Line 201: Remind the reader that the grey box refers to Figure 1.
- Line 212: "Trajectories was"
- Line 744: "$T_i\left( k\right)$ as the times that" should be "$T_i\left( k\right)$ as the number of times that".
- Line 810: $\widehat{\mu}_n = \frac{1}{n}\sum_1^n \widehat{\mu}_t$, what does this mean? Does it mean that you have $n$ samples from the optimal arm?
- Line 311: "we don't need" should be "we do not need".
- Line 349: "compatible to" should be "compatible with".
- Line 399: "an fair" should be "a fair".
- Line 423: "nexe" should be "next".
- Line 527: "we incorporates" should be "we incorporate".
- Line 535: "could broadening".
- Line 539: "for build".


I believe that the subject and the approach of the paper could be very interesting to investigate rigorously. Unfortunately, mainly due to the lack of clarity and the unsoundness of the theoretical results I have decided to reject the paper. Nevertheless, I invite the authors to take a look at my suggestions during the rebuttal period and maybe we can have an instructive discussion about a revised version of the paper.

**Questions:**

I have stated numerous questions in context in the weaknesses section. I would appreciate if the authors address them during the rebuttal phase.

---

> ### Author Response · Authors · 2024-11-20
> **Thank you for your high quality review**
>
> Dear Reviewer,
>
> First of all, I would like to express my deep gratitude for the time and effort you have put in. You gave very detailed and substantial review comments. I'll reply step by step in the following comments.

---

> ### Author Response · Authors · 2024-11-20
> **Regarding Algorithm 1**
>
> The functions in Algorithm 1 are simplified versions of our actual code. We've put the full code in an anonymous link(https://anonymous.4open.science/r/ReZero-D6AD/README.md). We will attach the open source code to the final version of the paper for easy reference.
>
> And we are sorry for the confusion caused by this simplified version of the code. You can find the corresponding functions(not strictly the same name) in https://anonymous.4open.science/r/ReZero-D6AD/lzero/mcts/buffer/game_buffer_rezero_mz.py . Specifically, the search is carried out in parallel in an entire batch. So prepare() do some centralized preprocessing of the data. origin_MCTS() is the original form of MCTS used in MuZero. traverse() refers to the forward process used in standard MCTS (from the root node to a leaf node). select_root_child() uses Equation 5 to select child nodes at the root node.
>
> We will add the above short description to the new version of the paper, hoping that pairing it with open source code will reduce the confusion of readers.
>
> In addition, reanalyze refers to a part of the MuZero algorithm which conducts MCTS on buffer datas. While traverse refers to the forward search process in MCTS (each forward search from the root node to a leaf node constitutes a traverse). So they're not on the same level. I hope this answer will reduce your confusion. Your further questions and suggestions are always welcome.

---

> ### Author Response · Authors · 2024-11-20
> **Regarding stochastic dynamics**
>
> We can understand your concerns. ReZero itself is a general improvement method for the MCTS+RL series of algorithms. What it can handle depends on what algorithm it is applied to. For example, applying ReZero to Stochastic MuZero(https://openreview.net/forum?id=X6D9bAHhBQ1) can speed up stochastic dynamics cases. Applying ReZero to Sampled MuZero(https://arxiv.org/abs/2104.06303) can speed up the case of continuous action space(As we showed in DMC control tasks. the baseline is actually Sampled MuZero in DMC tasks, and we made a mistake on legend). Here Equation 5 uses the case of MuZero as an example to make a general illustration.

---

> ### Author Response · Authors · 2024-11-21
> **Regarding calculation mistakes in (13) and (24)**
>
> After checking the calculation, I find that it should be $e^{2\frac{\sqrt{a}\varepsilon}{C^2}}$ in 13(and the final result in (14) is right).
>
> And it should be $1 + \frac{2C^6}{\varepsilon^4P_1^2}$ in (24). So I need to correct the constant term in the upper bound.
>
> I apologize for these mistakes, which may have been caused by negligence in the calculation or when typing the formula. Thank you for meticulously checking these calculations and helping me correct my mistakes!

---

> ### Author Response · Authors · 2024-11-21
> **Regarding the passage from (26) to (27)**
>
> Thank you for inviting me to discuss this. The scaling from (26) to (27) is inspired by the methods in bandit algorithms(https://tor-lattimore.com/downloads/book/book.pdf). You can refer to the second inequality in the last paragraph of the proof of Theorem 8.1 in Chapter 8 (page 120), which is an almost identical scaling.
>
> The inequality you provided uses a double summation, and in my understanding, this scaling is similar to the one I used in the step from (18) to (19). However, from step (26) to (27), I adopted a different scaling method.
> The insight behind this is as follows: In (26), if a certain $t$ activates the indicator function, which means
>
> $\mathbb{I}${$
> \hat{\mu}_i(t-1)+P_i \sqrt{\frac{n-1}{(1+T_i(t-1))^2}}\ge  \mu_1-\varepsilon \
>  \text{and}  \ A_t  = i
> $} $=1$
>
> it necessarily corresponds to a unique $s=T_i(t-1)$ that activates one of the indicator functions in the summation in (27), which means
>
> $\mathbb{I}$ {$\hat{\mu}_{is}+P_i \sqrt{\frac{n-1}{(1+s)^2}}\ge  \mu_1-\varepsilon$ } $=1$
>
>  Specifically, the condition $A_t=i$ plays a crucial role. Without this condition, it would not be guaranteed that different $t$s correspond to different $s$s. For example, suppose $t_1$ and $t_2$ each activate a indicator function in (26), but during this period, arm $i$ might not have been selected. In this case, $s_1=T_i(t_1-1)$ and $s_2=T_i(t_2-1)$ correspond to the same $s$, then simply summing over $s$ cannot guarantee an upper bound, since some certain $s$s may need to be considered more than one time. Therefore, a double summation is needed. However, under the condition $A_t=i$, once $t_1$ activates the indicator function, it means that the times arm $i$ being selected is also updated. If $t_2>t_1$ and $t_2$ also activates the indicator function, then there is always $s_2>s_1$, which means that each activated indicator function in (26) uniquely corresponds to an activated indicator function in (27), so (27) is an upper bound for (26).
>
> I hope this explanation meets your satisfaction, and further discussion is always welcome. I am very grateful for the references you provided, which may help broaden my perspective and understand MCTS from different angles.

---

> ### Author Response · Authors · 2024-11-21
> **Regarding Clarity**
>
> I can sense the effort you have put into understanding ReZero. I believe the confusion in Section 4.1 might stem from our differing background and domain knowledge. Our approach is an improvement upon the MuZero method, and the way we collect data, train networks, and so on, follows the MuZero framework. Therefore, we focus only on the improved modules and their corresponding analyses, without reintroducing the entire algorithm flow from scratch. This practice is in line with many well-regarded papers in the field: Sampled MuZero((https://arxiv.org/abs/2104.06303), Gumbel MuZero(https://openreview.net/forum?id=bERaNdoegnO), MuZero Unplugged(https://openreview.net/forum?id=HKtsGW-lNbw), and so on(apart from that, Stochastic MuZero(https://openreview.net/forum?id=X6D9bAHhBQ1) does have a short section on model training, as it makes changes to the network model itself).
>
> Specifically, ReZero mainly modifies the reanalyze module of the MuZero algorithm. We have altered batch organization and MCTS setting during reanalyze. We use the known sample mean as estimation of a specific child node (for the remaining child nodes, the UCB value is still used). This change occurs only when selecting child nodes for the root node. For non-root nodes, the method of selecting child nodes remains the same as the original MCTS method(i.e. using UCB values) in MuZero.
>
> Additionally, I noticed your concern about exploration. In our modified MCTS, there is indeed one child node that is directly evaluated using estimation. (This means that no exploration is conducted for this child node, but exploration for the remaining child nodes is maintained as it should be.) Therefore, I understand your concern that if this estimation has a large variance, it could affect the search. My explanation for this is that the variance of this estimation also decreases as the number of forward searches in MCTS increases. For example, if we set the number of forward searches in a single MCTS to $n$, since our estimation is reused from one MCTS(as showed in Figure 2), it is also obtained through $n$ samplings(For comparison, in the original MCTS setting, though we keep exploring this arm, the number of times we ultimately select this arm will not exceed $n$, because the total number of forward searches is $n$). Therefore, in our theoretical analysis, as $n$ increases, the regret can still maintain sublinear growth.
>
> In other words, as long as the provided estimation is sufficiently good, exploration is not necessary. The most extreme example is the one-armed bandit model(checck exercise 8.3 in page 121 of https://tor-lattimore.com/downloads/book/book.pdf), if the exact parameters are already known, there is no need to explore the corresponding arm. In our setting, the exact expected return is unknown, so we use an estimation obtained after sufficiently many samplings.
>
> I fully understand the confusion you have experienced, and I hope my response can help you understand our method on certain levels. Your further suggestions and comments are always welcome.

---

> ### Author Response · Authors · 2024-11-22
> **Regarding mini-batches and trajectories**
>
> In reinforcement learning, we often have a buffer to store data generated during interactions with the environment. The purpose of reanalyze is to update old data to ensure fresh training targets. When reanalyzing, we sample some data from the entire buffer to form a batch and do MCTS for every sample in it. This batch is collected in units of trajectories. Here, a trajectory is actually a segment cut out from a complete game. We fix the length of these segments and ensure they do not intersect (to avoid redundant searches and waste computational resources). Next, we further split the batch into several mini-batches, placing states with the same sequence number from each trajectory into the same mini-batch. The states in the same mini-batch are executed MCTS in parallel (this is merely a programming technique to allow the model to infer in parallel and save time). As shown in Figure 2, when reusing information, each state in the mini-batch will only receive the value transmitted from its own trajectory, meaning only one child node would be evaluated by estimation value, while the remaining child nodes need to be explored normally.
>
> I hope this answer can help reduce some of your confusion, and further discussion and questions are always welcome.

---

> ### Author Response · Authors · 2024-11-22
> **Regarding misspellings and notations**
>
> We have fixed the corresponding errors in the new version of our paper. Additionally, we have modified some notations and improved the statement of the theorem. Thank you very much for your correction and helping us improve the paper.

---

### Official Review · Reviewer_voYt · 2024-11-04

**Soundness:** 3
**Presentation:** 3
**Contribution:** 3
**Rating:** 6
**Confidence:** 2

**Summary:**

The paper proposes an improvement to the MuZero algorithm. Instead of trying the child node selection as a multi-armed bandit problem, the method treats it as one-armed bandit problem, that is a problem of selecting between one stochastic arm or a sure pay-off with a known value. Since this true value is unknown, of course, authors propose a backward search technique that produces an estimate of it.

**Strengths:**

- The paper is well-written and authors include easy-to-understand figures
- The authors propose a theoretical justification of their framework
- Presented experimental results clearly show the proposed method is faster (in term of wall-clock time) than MuZero

**Weaknesses:**

- There only baseline authors compare against is the MuZero algorithm. However, in the related work section authors mention many improvements proposed to the MuZero baseline to make it faster.

**Questions:**

- Why MuZero is the only baseline authors compare with?

---

> ### Author Response · Authors · 2024-11-20
> **Regarding the Baseline**
>
> Thank you for your efforts and nice comments  on our work! In addition to MuZero, our baseline also contains EffeicientZero (in the appendix) and Sampled MuZero (not explicitly stated in the legend due to our negligence. In fact, the algorithm compared in the two DMC control tasks is not MuZero, but Sampled MuZero, and we will fix it in new version). These comparisons are intended to demonstrate the versatility of ReZero in combination with different algorithms, like we mentioned in line 74: ”Therefore, our algorithm design is universal and can be easily applied to the MCTS-based algorithm family”.
>
> We understand your concerns. As you pointed out, We have mentioned some other acceleration methods in Section 2.2. We mentioned PTSAZero and SpeedyZero, both of which are orthogonal to ReZero in principle, so they can be seamlessly integrated with ReZero. In addition, we have also mentioned the difference between KataGo and our approach in Section 2.2. It is important to clarify that we do not claim that our backward-view reanalysis strategy is superior to these methods. Instead, we propose a novel perspective on information reuse from a backward-view approach. Therefore, at this stage, we did not directly compare the performance with other acceleration algorithms.

---

### Official Review · Reviewer_W74e · 2024-11-04

**Soundness:** 2
**Presentation:** 1
**Contribution:** 1
**Rating:** 3
**Confidence:** 2

**Summary:**

In MuZero, the "reanalyze" mechanism enhances sample efficiency by revisiting and updating past experiences stored in the replay buffer. In this paper, the authors propose a method to use information from future time steps during the reanalyze phase to reduce the search space and accelerate individual MCTS runs.

**Strengths:**

ReZero demonstrates a reduction in wall-clock time needed to achieve comparable performance levels compared to baseline MCTS-based algorithms.

**Weaknesses:**

- The writing is not clear enough for the reader to understand exactly what the algorithm is doing. I had a hard time understanding Section 4.1, and Figure 2, so I am not able to evaluate this paper too well.
- To my best guess, the authors are proposing to use previous MCTS runs to approximate the value of nodes in future MCTS runs. This is based on the
- I don't think Theorem 1, which seems to be based on existing work, is properly grounded in the current setting. This is because MCTS is not a typical bandit: I recommend the authors take a look at the Shah et al. paper "Non-asymptotic analysis of monte carlo tree search".

**Questions:**

See weaknesses.

---

### Author Response · Authors · 2024-11-22

Dear reviewers,

Thank you for your recognition of this work, encouragement for our attempts, and constructive criticism of our shortcomings. We have uploaded a new version and highlighted the modified parts in blue. We have corrected some misspellings, added some annotations, and improved the statement of theorem 1 and its proof. Additionally, we have prepared an anonymous link(https://anonymous.4open.science/r/ReZero-D6AD/README.md) to share our code.

One issue that needs a unified response is that the baselines we compare actually include MuZero, Sampled MuZero, and EfficientZero(in appendix). Due to our previous oversight, they were not correctly labeled in the legend. We have corrected this in the new version. Due to time constraints, we have done our best to respond to everyone's suggestions and questions. If there are any omissions or further questions,  welcome your new replies.

---

### Author Response · Authors · 2024-11-26

Dear Reviewers,

We would like to express our sincere gratitude for the time and effort you have invested in reviewing our paper. During the rebuttal phase, we've responded to some key issues. We look forward to your comments before the extended deadline. We are deeply grateful for the time and effort cost that may be incurred.

Sincerely

Authors

---

### Meta-Review · Area_Chair_LCzB · 2024-12-22

**Metareview:**

Th paper tackles the problem of improving the sample efficiency of MCTS, with some finite time guarantees.
The reviewers found several weaknesses with the current paper.
In particular, they the writing seems to be poor, as highlighted by two reviewers. There also seems to be some mistakes in the proof. Finally, the empirical evaluation may not be very thorough because it is not comprehensive in terms of baselines.

**Additional Comments On Reviewer Discussion:**

There was no substantial discussion as the reviewers overall felt that the paper is not ready for publication

---

### Decision · Program_Chairs · 2025-01-22

Reject